# Targeting the Inflammatory Hallmarks of Obesity-Associated Osteoarthritis: Towards Nutraceutical-Oriented Preventive and Complementary Therapeutic Strategies Based on n-3 Polyunsaturated Fatty Acids

**DOI:** 10.3390/ijms24119340

**Published:** 2023-05-26

**Authors:** Laura Gambari, Antonella Cellamare, Francesco Grassi, Brunella Grigolo, Alessandro Panciera, Alberto Ruffilli, Cesare Faldini, Giovanna Desando

**Affiliations:** 1Laboratorio Ramses, IRCCS Istituto Ortopedico Rizzoli, via di Barbiano 1/10, 40136 Bologna, Italy; antonella.cellamare@ior.it (A.C.); francesco.grassi@ior.it (F.G.); brunella.grigolo@ior.it (B.G.); giovanna.desando@ior.it (G.D.); 21st Orthopedic and Traumatology Clinic, IRCCS Istituto Ortopedico Rizzoli, via G.C. Pupilli 1, 40136 Bologna, Italy; alessandro.panciera@ior.it (A.P.); alberto.ruffilli@ior.it (A.R.); cesare.faldini@ior.it (C.F.)

**Keywords:** osteoarthritis, obesity, inflammation, pain, dysbiosis, phytochemicals, nutraceuticals, dietary supplementation, n-3 polyunsaturated fatty acids, translational interventions, intra-articular deliveries

## Abstract

Obesity (Ob), which has dramatically increased in the last decade, is one of the main risk factors that contribute to the incidence and progression of osteoarthritis (OA). Targeting the characteristics of obesity-associated osteoarthritis (ObOA) may offer new chances for precision medicine strategies in this patient cohort. First, this review outlines how the medical perspective of ObOA has shifted from a focus on biomechanics to the significant contribution of inflammation, mainly mediated by changes in the adipose tissue metabolism through the release of adipokines and the modification of fatty acid (FA) compositions in joint tissues. Preclinical and clinical studies on n-3 polyunsaturated FAs (PUFAs) are critically reviewed to outline the strengths and weaknesses of n-3 PUFAs’ role in alleviating inflammatory, catabolic and painful processes. Emphasis is placed on potential preventive and therapeutic nutritional strategies based on n-3 PUFAs, with a focus on ObOA patients who could specifically benefit from reformulating the dietary composition of FAs towards a protective phenotype. Finally, tissue engineering approaches that involve the delivery of n-3 PUFAs directly into the joint are explored to address the perspectives and current limitations, such as safety and stability issues, for implementing preventive and therapeutic strategies based on dietary compounds in ObOA patients.

## 1. Introduction

Obesity (Ob) is one of the main risk factors for osteoarthritis (OA), and its prevalence has dramatically increased worldwide [1,2,3,4], thus making obesity-associated OA (ObOA) a significant burden on the healthcare system, involving social, biological and psychosocial aspects. A sedentary lifestyle and a high-caloric diet are the principal drivers of Ob, rendering patients with a body mass index (BMI) >30 kg/m^2^ 6.8 times more susceptible to the development of OA [5]. The increased adiposity and body weight in obese patients is classically associated with biomechanical alterations in joint tissues and synovial fluid (SF) components [6], resulting in mechanical tissue damage. However, the growing evidence of OA in non-weight-bearing joints of obese patients has boosted the identification of alternative causative mechanisms related to altered lipid metabolism homeostasis, causing low-grade inflammation in all joint tissues. Adipose tissue acts as an endocrine organ through the release of soluble mediators, including cytokines, adipokines and specific forms of fatty acids (FAs), and it induces detrimental effects on joint tissues [7]. In particular, most adipokines (i.e., leptin, visfatin and resistin) promote the activation of an inflammatory circuit, known as meta-inflammation, which leads to the synthesis of degradative enzymes (i.e., aggrecanases and metalloproteinases), reactive oxygen species (ROS) and prostaglandins [8,9]. In OA joints characterized by a reduced viability of chondrocytes and erosion processes, the accumulation of ectopic lipids, such as arachidonic acid, occurs, indicating a pivotal role of those FAs in OA pathophysiology [10]. Moreover, Ob-associated oxidative stress causes the lipolysis of adipocytes and increases concentrations of free FAs in the bloodstream and tissue, activating the pro-inflammatory M1 macrophage population. In this direction, various studies have explored the effects of an FAs-enriched diet on articular joints, showing a tight correlation between inflammation and an increased n-6/n-3 polyunsaturated FAs (PUFAs) ratio [11,12,13]. In particular, a high dietary intake of n-6 PUFAs in obese patients has been shown to promote inflammation, which causes synovitis and cartilage degradation [14,15]. Additionally, these patients show low plasma concentrations of n-3 PUFAs [16], which play a protective role in inflammation-based disorders, including OA [17,18].

Overall, changes in the lipidic composition in cartilage and the activation of inflammation in obese patients lead to the onset of ObOA and drive the evolution towards a more severe form of OA. Dietary interventions using natural food compounds (nutraceuticals) have emerged as a promising, alternative and complementary option to conventional therapy with limited side effects [19,20].

The most common recommendations include a diet rich in fruits and vegetables due to their antioxidant and anti-inflammatory effects. Given the altered ratio of n-3/n-6 PUFAs in patients with ObOA, reformulating their dietary supplementation in favor of n-3 PUFAs could pave the way for new preventive and therapeutic strategies for the management of metabolic OA in this patient cohort [21]. However, bioavailability studies have highlighted some pitfalls of oral PUFA supplementation, urging the scientific community to identify new delivery systems to increase their amount in the blood and within the damaged joint [22]. This review aims to evaluate the role of n-3 PUFAs in the joint protection of patients with ObOA, focusing on critical inflammatory aspects that severely disrupt key checkpoints in joint tissues. It also describes the current state-of-the-art and potential new opportunities to increase their bioavailability, with a look at potential clinical perspectives for personalized strategies in this patient cohort, including intra-articular (IA) administration.

### Literature Search Strategy

Scopus and PubMed literature searches were performed with the following keywords: “osteoarthritis” “inflammation” “obesity”, “dysbiosis”, “n-3 polyunsaturated fatty acids (n-3PUFAs)”, “eicosapentaenoic acid (EPA)”, “docosahexaenoic acid (DHA)”, “resolvins”, “maresins” and “intra-articular delivery”.

## 2. Pathogenesis of ObOA: From a Biomechanics-Centered Theoretical Paradigm toward a Key Role of Inflammation

First indications of Ob as an independent risk factor of OA [23,24] date back to 1988, when the Framingham study, performed in a cohort study of patients with knee OA, showed a link between Ob and OA [25]. OA was traditionally considered a wear-and-tear disease involving cartilage tissue. until several shreds of evidence emerged on the key contribution of all joint cells and other key processes beyond cartilage degradation, such as inflammation [26,27]. Similarly, ObOA’s etiology is now recognized as a complex multifactorial process, in which, besides wear factors, systemic and chronic low-grade inflammation plays a central role in its onset and progression (Figure 1) [26,27]. Figure 1 summarizes the new paradigm of ObOA’s etiology, which is further described in the following paragraphs.

### 2.1. Role of Impaired Loading and Biomechanics in Joint Degeneration in ObOA

Joint degeneration in ObOA pathophysiology has been primarily attributed to changes in mechanical loading as a result of increased body weight [26,28]. The mechanobiology of joint tissues and the lubricating role of SF play pivotal roles in governing joint biomechanics through mechanotransduction [29]. Several in vitro studies have shown that mechanical forces modulate the expression of proteins belonging to the extracellular matrix (ECM) of cartilage, comprising type II collagen (Coll II) and aggrecan [30,31,32], which are crucial for ensuring the proper biomechanical properties of articular cartilage. In particular, Coll II is known to withstand shear and tensile forces in articular cartilage, while proteoglycans such as aggrecan and glycosaminoglycans (GAGs) provide compressive resistance [33]. Interestingly, the surface of the cartilage is covered with a layer of phospholipids that serves as a boundary lubricant during joint loading [34]. Impaired biomechanics, also involved in the onset of post-traumatic OA [3,35], dysregulate the biomechanical pathways governing joint metabolism, leading to the synthesis of inflammatory and catabolic mediators, including metalloproteinases (MMPs) and aggrecanases (ADAMTs), which affect cartilage components [36]. Ob often leads to an abnormal and unequal distribution of mechanical loading, which evolves into progressive wear-and-tear of the joint [37]. Because of its anatomical configuration, the knee joint is the most affected, it being a hinge joint in which stability is determined by the integration of various surrounding tissues (synovium, ligaments, cartilage, bone) [38]. Moreover, ObOA alters the tribological properties of articular cartilage by another mechanism; dietary intake of FAs in obese patients drives changes in the phospholipid layer lining of the articular surface, leading to the alteration of this boundary lubricant during joint loading [39,40]. These results demonstrated how dietary FAs are determinants of the onset of OA altered biomechanics beyond altered joint loading in ObOA.

SF, also known as joint fluid, exerts a critical role in joint biomechanics by protecting joints from impact, friction and disease. The synovium and infrapatellar fat pad (IFP) play vital roles in the progression of ObOA since they are the main tissues for the secretion of joint fluid. The biochemical composition of SF ensures its biomechanical properties due to the presence of proteins and molecules secreted by synovial cells and IFP, including the lubricating molecules hyaluronan (HA) and proteoglycan 4 (also called lubricin), phospholipid species (PLs), cytokines and metabolic by-products. In addition, SF plays a metabolic role by nourishing the avascular articular cartilage and modulating synovial inflammation [36,38,41,42,43]. The SF of OA patients shows altered rheological properties, which largely depend on the amount and concentration of HA (lowered HA concentration with a shift to low-molecular-weight forms) [44] and the extent of inflammation [45]. Along with HA, changes in lubricin and PL contents cause an altered lubrication of the joint [40,46]. In addition, inflammation of the synovial membrane impairs its ability to filter molecules, leading to an increase in the amount and concentration of proteins and a decrease in SF viscosity [47]. Besides these changes, obese patients show an alteration of SF components for the presence of inflammatory molecules, including adipokines, which are released from the excessive adipose tissue accumulated in the synovium and IFP or from the blood system [48,49].

Overall, altered joint biomechanics ultimately activate inflammatory pathways and cause pain, limiting daily activities with a huge socioeconomic impact. The restoration of the altered joint biomechanics is highly demanded among OA patients and ObOA to recover their daily and work activities. Therefore, the development of selected treatments should be targeted toward the promotion of cartilage regeneration and restoration of SF viscosity within the joint to withstand external stimuli (shear stress, compressive forces, etc.) during joint loading, as well as by acting on several mechanical and molecular mediators beyond mechanotransduction [40].

### 2.2. Role of Inflammation in Joint Degeneration in ObOA

The identification of OA in non-weight-bearing joints, such as the hand and wrist, sheds light on the presence of other causative mechanisms, along with biomechanical factors, behind the pathophysiology of ObOA. The impact of chronic inflammation and metabolic state in obese patients has become increasingly more evident [24,50]. In this light, the role of adipose tissue as an endocrine organ in mediating the systemic effects of ObOA via the release of inflammatory mediators such as adipokines and FAs has been postulated, thus offering new prospects for interventions targeting the adipose-inflammatory aspects of this metabolic disorder [51]. In support of this hypothesis has emerged the presence of synovial inflammation and high concentrations of adipokines in SF and plasma among clinical manifestations in ObOA patients [48,49]. Interestingly, structural and metabolic alterations occur not only in the visceral adipose tissue but also in the IFP, which synergistically cooperate and release inflammatory mediators [52,53]. Several biological mechanisms have been proposed beyond inflammation in ObOA, even if specific pathways are currently not well understood [54].

First, in obese patients, adipose tissue mediates the release of cytokines (e.g., IL-1, IL-6, IL-8, tumor necrosis factor (TNF)-α) and adipokines (e.g., leptin, adiponectin, resistin, omentin, vaspin) that exert deleterious effects on joint tissues [48,51]. Clinical evidence of this inflammatory state in the SF of ObOA patients comes from Pearson MJ et al. who demonstrated that IL-6 levels are positively correlated with the levels of some adipokines [55]. Various signaling pathways intervene to feed this inflammatory network, including AMP-activated protein kinase (AMPK)/mammalian target of rapamycin (mTOR), nuclear factor-κB (NF-κB), mitogen-activated phosphokinase (MAPK) and protein kinase C (PKC). Besides inflammatory cytokines and adipokines, inflammation is further fueled by increased local and systemic concentrations of specific forms of FAs in obese patients due to their dietary intake (increased saturated fatty acids (SFAs), n-3/n-6 PUFAs ratio shifted in favor of the latter) [56] or to the oxidative-stress-induced lipolysis of adipocytes [57]. An imbalance in the FA composition increases pro-inflammatory lipids and reduces protective mediators (such as protectin D1), especially in adipose tissue, which has been linked to increased pain and impaired joint function [58]. Ob-induced oxidative stress also accelerates lipid peroxidation and causes protein and DNA alterations and increases the production of oxylipins [59] in the SF.

Interestingly, some authors have recently reported the scientific relevance of pyroptosis, a type of cell death that induces a strong inflammatory response in OA through the disruption of the cell membrane’s integrity, the release of cellular contents and inflammatory factors [60]. Two pathways are specifically involved in the activation of pyroptosis: TLR-4/NF-κB signaling and NOD-like receptor protein (NLRP3) inflammasome/caspase-1/gansdermin D (GSDMD) signaling, which serve as sensors for damage-associated molecular patterns (DAMPs) and pathogen-associated molecular patterns (PAMPs) and activate the innate and adaptative immune system, exacerbating inflammatory response [60]. Recently, the TLR4 signaling pathway has been identified as the main trigger of Ob-induced inflammation [61] and of ObOA [62]. However, its precise role in ObOA is still largely unknown [63,64].

Notably, G-protein-coupled receptor 120 (GPR120) [65], a receptor for n-3 PUFA, whose activation is associated with a reduced degree of OA severity, has been found to be down-regulated in OA and is inhibited by the infiltration of macrophages within adipose tissue [66]. It is not yet known whether GPR120 is modulated during ObOA; however, it is conceivable that it may further aggravate the consequences of altered FA status.

Finally, the activation of transient receptor potential vanilloid 1 (TRPV1), a cation channel, has been identified as one of the molecular mechanisms linking OA pain to metabolic disorders [67]. TRPV1 agonist (among which endogenous PUFAs e.g., anandamide, arachidonic acid and epoxyeicosatrienoic acids activate TRPV4 channels) provides a significant reduction in pain in patients with OA [68] thanks to its anti-inflammatory effects mediated by the activation of nuclear factor erythroid-2-related factor 2 (Nrf-2), the master regulator of anti-oxidant and anti-inflammatory responses [69]. The TRPV1 receptor is also involved in the inflammation of adipose tissue [70] and could be effective as a weight-lowering agent [71]. However, controversy exists in terms of its role in high-fat-diet (HFD)-induced Ob [72].

Second, the recruitment and infiltration of immune cells into the IFP and synovium is another typical feature of patients with ObOA [73,74]. Particularly, obese patients show enhanced synovitis and macrophage infiltration in the synovium accompanied by dominant M1 macrophage polarization [75,76]. Similarly, the increased activity of M1 in the IFP is caused by a combination of the HFD effect and OA [77] and contributes to changes in the integrity of the IFP. Moreover, in ObOA, there is an increased production of Th1, Th17, T-helper and B cells and a reduced number of Treg and NK cells [78,79]. Neutrophils, macrophages, monocytes and dendritic cells recognize PAMPs through pattern recognition receptors (PRRs) or DAMPs, which are increased in OA [80], driving a rapid-onset inflammatory response [78].

Third, alterations in the gut microbiota (dysbiosis) are associated with systemic inflammation in ObOA patients. Injuries to the intestinal mucosa caused by HFD and western-type diets, typical of Ob and metabolic syndromes, facilitate the passage of microbiota bacteria from the gut to the joint through the blood and increase the systemic circulation of lipopolysaccharide (LPS), which causes metabolic endotoxemia [81,82] and an increased accumulation in SF. The composition of the gastrointestinal (GI) microbiota and/or the production of microbial metabolites modulated by the diet may alter the pool of PAMPs [83]. Interestingly, a diet rich in different FAs also contributes differently to the development of endotoxemia by the direct stimulation of TLR4 by SFAs, acting as non-microbial agonists. In contrast, a Mediterranean diet containing higher amounts of PUFA-rich foods [84] reduces LPS toxicity, one of the key pathways involved in ObOA [62]. Moreover, ObOA is associated with reduced bacterial diversity, an altered presentation of bacterial genes and changes in metabolic pathways [85,86]; changes in microbe-associated lipid metabolites differently activate the innate immune system in ObOA [87]. An HFD contributes to an increase in the number of harmful bacterial species in the gut microbiome, leading to an increased inflammatory condition [88], while a fiber-rich diets are beneficial for health by positively modulating the composition of the microbiota [89,90]. (Figure 2). Interestingly, a slower digestion and/or the absorption of fats rich in PUFAs have been postulated as some of the reasons for the observed increased PUFA levels in obese patients [84].

Altogether, these aspects cause a marked joint degeneration, challenging clinicians’ choices toward approaches aimed at extinguishing this inflammatory picture [24], among which dietary recommendations are a valid alternative option.

## 3. State-of-the-Art of n-3 PUFAs’ Role in Counteracting Inflammation: Toward Evidence-Based Medicine for Their Use in ObOA

### 3.1. FAs Metabolism: Focus on Inflammation

Nutrient excess leading to Ob results in increased FAs in the circulation and the ectopic accumulation of FAs in joint tissues, triggering lipotoxicity and the onset or aggravation of OA [91]. However, recent evidence has supported the concept that these pathological consequences are not merely caused by excess fat intake but also that the type of fat intake is important. In this context, different types of FAs have been shown to exert distinct effects on inflammation [91]. According to their chemical structure, FAs can be categorized as (1) SFAs without double bonds, (2) monounsaturated FAs (MUFAs) with one double bond between their carbon molecules, and (3) PUFAs with more than two double bonds [92]. SFAs include butyric acid (contained in milk), caprylic acid/capric acid (contained in coconuts, milk, peanut butter and palm fruit oil), palmitic acid and stearic acid (contained in mammalian animal fats, including lard). Common MUFAs include oleic acid (contained in plant-based oils, including olive and avocado oil), palmitoleic acid (contained in macadamia nuts and oil, sea buckthorn oil and berries, avocado and olive oil) and vaccenic acid (contained in the fat of ruminants, milk, butter and yogurt).

PUFAs are FAs that consist of a hydrocarbon chain with a carboxyl (-COOH) and methyl (-CH3) group and are classified into omega-3 (ω-3 or n-3) and omega-6 (ω-6 or n-6) according to the position occupied by the first double bond at the terminal methyl-end of the molecule. Dietary linoleic acid (LA, n-6 PUFAs) and alpha-linolenic acid (ALA, n-3 PUFAs) are the simplest members of each family of PUFA. The human body lacks endogenous enzymes that promote the double bond formation of PUFAs, making PUFAs “essential” nutrients that must be taken [93] by including PUFA-rich nutrients into the diet [93]. LA is found in plant oils (sunflower, safflower and corn oils), cereals, animal fat and wholegrain bread; ALA is found in green leafy vegetables, flaxseed and rapeseed oils. Other sources of PUFAs include walnuts, sesame seeds, peanut butter and peanuts, poppy seeds and oils of avocado, olive, cod liver oil, *krill* oil, and *Antarctic krill (Euphausia superba)* oil [93].

LA and ALA are both metabolized by the actions of elongases and desaturases on downstream key lipid mediators, among which are arachidonic acid (AA, n-6 PUFAs), docosahexaenoic acid (DHA, n-3 PUFAs) and eicosapentaenoic acid (EPA, n-3 PUFAs) (Figure 3). The liver is the primary site for PUFA metabolism; however, it can also take place in various other tissues [94]. Notably, desaturase activity declines with age, leading to a deficiency in gamma-linolenic acid (GLA, n-6 PUFAs), di-homo-γ-linolenic acid (DGLA, n-6 PUFAs), AA, EPA and DHA and their metabolites in aged cells [95]. PUFAs are mainly metabolized to prostaglandin, thromboxanes and leukotrienes through three enzymatic pathways; they act as substrates of cyclooxygenases COX (COX-1 constitutive, COX-2 inducible), lipoxygenases (LOX) and cytochrome P450 (Figure 3) [59]. Particularly, AA (n-6 PUFAs) is converted by COX to prostaglandin H2 (PGH2) and further converted by a specific synthase to prostaglandins (prostaglandin E2 (PGE2), prostaglandin D2 (PGD2), prostaglandin I2 (PGI2) and prostaglandin F2α (PGF2α)), thromboxanes such as thromboxane A2 and leukotrienes such as leukotriene B4, while EPA (n-3 PUFAs) is converted to prostaglandin E3, thromboxane A3, prostaglandin I3 and leukotriene B5. The LOX pathway produces leukotrienes, 5-hydroxyeicosatetraenoic acid (HETEs) from AA and 9-hydroxy octadecadienoic acid (or 9-HODE) from LA. The cytochrome 450 pathway produces epoxyeicosatrienoic acids (EETs), dihy-droxyeicosatrienoic acids (DHETs) and HETEs from AA and epoxyoctadecamonoenoic acids and dihydroxyocta-decenoic acid from LA [59]. Moreover, both n-3 and n-6 PUFAs serve as precursors of endocannabinoids [96,97]. In particular, the non-oxidative pathways of DHA and EPA (n-3 PUFAs) generate docosahexaenoyl ethanolamide (DHEA) and eicosapentaenoyl ethanolamide (EPEA) via the N-acyl ethanolamine synthesis pathway, whereas arachidonic acid (AA)(n-6 PUFAs) is converted to ethanolamine (AEA, anandamide). DHEA, EPEA and AEA have similar effects to Δ9-tetrahydrocannabinol (THC), the active ingredient of cannabis sativa, by activating the cannabinoid receptors CB1 and CB2.

Studies investigating the impact of specific FAs on inflammation have grown considerably and have evidenced that different products of FA metabolism exert opposite effects on inflammation.

SFAs trigger an inflammatory phenotype by increasing circulatory levels of TNF-α, IL-6, COX-2, Nos2 and the activation of TRL-4 [98,99,100]. SFAs activate caspase-4/5 in human monocytes, triggering IL-1β and IL-18 release [101].

Similarly, n-6 PUFAs are often associated with pro-inflammatory effects through the promotion of the synthesis of eicosanoids such as prostaglandins (PGE2, thromboxane A2 and PGI2) and leukotrienes (leukotriene B4, 5-hydroxyeicosatetraenoic acid and 9-HODE) [58], which are also termed oxylipins, whose formation is also induced by ROS-dependent lipid peroxidation. The activation of the TLR-4 by n-6 PUFAs may further feed the inflammatory circuit by promoting (1) the release of cytokines through the NF-KB signaling pathway [60] and (2) the induction of pyroptosis, a programmed cell death, through regulation of the NLRP-3 inflammasome [60,102]. However, some contradictory results have been published [103] on the role of n-6 PUFAs in inflammation.

On the other hand, MUFAs such as AEA promote beneficial anti-inflammatory mechanisms (M2 macrophage polarization, adipocyte IL-10 secretion and the inhibition of NLRP3 inflammasome) and reverse the deleterious effects of SFAs on adipose tissues [98]. Most importantly, n-3 PUFAs have been shown to exert anti-inflammatory and pro-resolving properties [104,105,106]. The competitive inhibition with n-6 PUFAs during metabolism represents one of the first primary theories formulated to explain why n-3 PUFAs elicit therapeutic anti-inflammatory action [107]. Notably, oxylipins generated from n-3 PUFAs exert an anti-inflammatory phenotype and have been termed as pro-resolving oxylipins or specialized pro-resolving mediators (SPMs), e.g., resolvins (RvD), protectins (PDX), maresins (Mar-1) and lipoxins.

### 3.2. Effect of n-3 PUFAs in ObOA: Focus on Anti-Inflammatory and Pro-Resolving Mediators

Strong evidence of the protective potential of various n-3 PUFAs metabolites (ALA, EPA and DHA, as well as SPMs) in OA comes from several in vitro preclinical investigations, as summarized in Table 1 [108]. Most of these studies attempted to mimic the pathophysiological state of OA using approaches based on joint cell cultures (e.g., chondrocytes and synoviocytes) or tissue explants from OA patients treated with exogenous inflammatory stimuli (IL-1β, TNF-α, IL-α, sodium nitroprusside-SNP and LPS). These models recapitulate several alterations in OA, including (i) increased inflammatory mediators (IL-1 β, TNF-α, COX-2, etc.); (ii) the low synthesis of typical ECM molecules (Coll II, proteoglycans, sulphated GAG, HA and aggrecan) and inhibitors of matrix degradative enzymes such as tissue inhibitors of metalloproteinases (TIMPs); and (iii) an increased presence of catabolic mediators (collagen I and X and proteases such as MMPs and ADAMTSs) [109,110]. Notably, studies on articular chondrocytes of both human and animal species have shown the effect of n-3 PUFAs (DHA, EPA, ALA, RvD1 and PDX) in reversing the inflammation-induced up-regulation of catabolic mediators [65,111,112,113,114,115,116] and in fostering the synthesis of anabolic markers [60,111].

In this line, Jin et. Al. showed that DHA exerts its anti-inflammatory action by the direct regulation of TLR4/NF-κB and the NLRP3/caspase-1/GSDMD signaling pathways through the inhibition of pyroptosis in chondrocytes [60]. This aspect holds great relevance in the context of ObOA, as both pathways are critically linked to ObOA-induced inflammatory response [61]. Interestingly, the same study compared the effects of n-3 PUFAs on SFAs (palmitic acid—PA), MUFAs (oleic acid—OLA) and n-6 PUFAs (LA), and it determined that the usage of SFAs and n-6 PUFAs increased TLR4/NF-κB signaling and the NLRP3/caspase-1/GSDMD signaling pathway, while treatments with n-3 PUFAs and MUFAs had an inhibitory effect [60]. These findings corroborated the pro-inflammatory effect of SFAs and n-6 PUFAs and the anti-inflammatory effect of MUFAs and n-3 PUFAs. Further indications of DHA’s protective role derived from studies conducted on a marine green-lipped mussel (GLM) extract, which contains abundant DHA and many minor fatty acids (including 5-, 9-, 12- and 16-nonadecatetraenoic acid, 5-, 9-, 12- and 15-octadecatetraenoic acid and 5-, 9-, 12-, 15- and 18-heneicosapentaenoic acid). GLM has shown optimal anti-inflammatory and antioxidant joint-protective properties, particularly by reducing levels of inducible nitric oxide synthase (iNOS), which is implicated in the development of oxidative stress during OA. Moreover, GLM has been demonstrated to inhibit necroptosis [115], a recently recognized form of cell death that elicits a more robust inflammation response than apoptosis through the release of DAMPs and the subsequent breakdown of the ECM. Necroptosis is mainly mediated by the action of receptor-interacting protein kinase 1 and 3 (RIPK-1, -3) and mixed-lineage kinase domain-like protein (MLKL) [116], which have been shown to be inhibited by GLM in OA patients.

Other authors have identified inflammatory pathways modulated by DHA, particularly Wang et al., who demonstrated the inhibition of p38 MAPK signaling [106], and Chen et al., who demonstrated the inhibition of Wnt/β-catenin and NF-κB signaling pathways [65]. Feng et al. provided molecular insight beyond the inhibition of Wnt/β-catenin and NF-κB signaling, demonstrating how this is reduced by DHA treatment through the down-regulation of a pivotal long non-coding RNA (lncRNAs) mediator, namely metastasis-associated lung adenocarcinoma transcript 1 (Malat1) [111]. Experimental evidence of the beneficial effects of DHA also comes from its metabolites, which include RvD [117] and PDX [118]. Benabdoune et al. found that RvD1 can reduce inflammation and catabolism by inhibiting NFkB/p65, p38 MAPK and JNK1/2 in human OA chondrocytes [117]. Piao et al. showed that PDX exerts a strong protective effect against the IL1β-mediated inflammatory response in rat chondrocytes by reversing the increased expression of MMPs and inflammatory cytokines and restoring elevated Coll II expression through the activation of AMPK and the inhibition of NF-κB/IκB, a pivotal inflammatory pathway in OA [118]. Notably, this was the first study investigating mechanisms upstream of NF-κB inactivation, and it found that the activation of AMPK signaling pathway by PDX is key to the downstream suppression of the activation of NF-κB [118]. These studies are of particular interest given that they have shown that SPMs, which are usually produced after the activation of host cells during infection, are able to restrain inflammation, thus resolving the infection [119].

Other studies have identified the effect of exogenous stimulation with EPA in chondrocytes, highlighting the fact that it exerts effects on (i) inflammation and catabolism processes by reducing several cytokines (PGE2, COX-2, IL-1α, IL-1 β, TNF- α and iNOS) and proteases (MMP-3, -13 and ADAMTS-4/-5), (ii) apoptosis through the downregulation of caspase 3 and poly (ADP-ribose) polymerase cleavage and (iii) anabolic metabolisms through increased GAGs [112,113,114]. Some studies have compared the efficacy of several n-3 PUFAs on inflamed chondrocytes [112,120] or cartilage explants [113]. ALA is thought to be less effective than EPA and DHA due to its rapid β-oxidation and its poor conversion to EPA and DHA. To provide a better understanding of this aspect on the cellular level, Zainal et al. compared the efficacy of ALA, EPA and DHA in bovine chondrocytes, demonstrating that EPA was the most effective at achieving multiple targets with lower concentrations, while DHA and ALA were less effective [112]. Given that the rate of conversion of ALA to EPA and DHA is low, these findings show the importance of direct supplementation with EPA and DHA. Notably, Yu et al. tested the effect of different ratios of n-6 PUFA/n-3PUFA (LA:ALA) in high-density chondrocyte cultures. Interestingly, an LA:ALA ratio of 1:1 was the most effective in reducing the catabolic effect compared to ratios of 2:1, 4:1, 6:1, 8:1 and 10:1 [121]. This finding is of clinical interest in ObOA, where an altered n-6 PUFA/n-3 PUFA ratio at the tissue level has also been documented. Likewise, Adler et al. found EPA to be more efficacious than DHA [120] in controlling inflammation and catabolic processes, while Wann et al. showed that EPA had a better GAG loss reduction capability than DHA in the long term, although their efficiencies were similar initially [113]. Through a direct comparison of COX-2 and COX-1 levels, Zainal et al. showed that ALA, EPA and DHA downregulate COX-2 while not affecting COX-1 [112]. COX-1 and COX-2 are two structurally distinct forms of the cyclo-oxygenase enzyme; COX-1 is a constitutive member of normal cells and COX-2 is induced in inflammatory cells. The inhibition of COX-2 activity represents the most likely mechanism of action for NSAID-mediated analgesia, while adverse effects have been attributed to COX-1 inhibition. In this light, Zainal et al. hypothesized that n-3 PUFAs give better results than NSAIDs based on the better COX-2/COX-1 inhibitory ratio [112], thus opening new insights into their potential use in combination with therapeutic interventions to reduce parent drugs.

Besides studies on chondrocytes, further investigations were carried out to assess the impact of n-3 PUFAs on synovial fibroblasts of both human and animal species. Similar to chondrocytes, the treatment of inflamed synoviocytes with n-3 PUFAs determined a decrease in catabolic and inflammatory mediators [108]. Interestingly, Caron et. al. identified that the lipidic composition of synoviocytes membranes is influenced by diet, which, in turn, influences the profile of downstream lipid mediators and signaling processes during inflammation [108]. Indeed, EPA and DHA, incorporated on synovial membranes, counteracted IL-β-induced inflammation by generating several pro-resolving docosanoids (RvD1, RvD2, Mar and PDX) [108]. However, in another study, similar SPMs (RvD1, RvE1 and MaR1) were not able to modulate inflammatory and catabolic markers induced by stimulation with TNF-α [122]. The elucidation of the molecular mechanisms implicated in the protective effects of n-3 PUFAs on inflammatory synoviocytes comes from two preclinical in vitro studies that identified the activation of Hippo-YAP [123] and the inhibition of the NF-κB p65 pathway [124]. Interestingly, Su et. al. showed that RvD1 activates anti-inflammatory molecules in synoviocytes by activating Hippo-YAP signalling pathway, thus leading to YAP cytoplasmic sequestration and proteasomal degradation [123]. Notably, this is of great interest in the context of OA, given that the inhibition of YAP has been proposed as a mechanism to prevent OA development [125], and in Ob, given that it is increased in response to these mechanical changes in hypertrophic adipocytes [126].

Lu J et al. demonstrated that MaR-1 suppresses IL-1β-induced MMP-13 secretion via the activation of the PI3K/AKT pathway and the inhibition of the inflammatory NF-κB pathway [124]. 

Besides the effects on synovial fibroblasts, some authors have explored the effect of n-3 PUFAs on immune cells, including T cells and macrophages. Notably, de Bus I et al. compared the biological effects of two endocannabinoids from DHA on LPS-stimulated RAW 264.7 macrophages, demonstrating a more pronounced response for DHEA than 13- and 16-H-DHEA [127]. Further in-depth studies were addressed to assess the potential of n-3 PUFA compounds in modulating the macrophage phenotype towards a more protective phenotype, considering their high plasticity and the prominent inflammatory M1 subset in Ob patients. To this regard, Dalli J et al. evaluated a precursor of MaR-1, 13S,14S-epoxy-DHA, demonstrating its role in modulating the polarization of M1 towards a pro-resolving M2 phenotype, and novel anti-inflammatory mechanisms through (i) the inhibition of leukotrien B4 and (ii) reduced arachidonic acid conversion [128].

To date, no in vitro models can mimic Ob-induced pathological changes. However, the above-described in vitro models of inflammatory OA have been useful in gaining insights into the PUFA-mediated molecular mechanisms in ObOA and into new biological targets. Recently, some in vitro studies have attempted to investigate the role of chronic low-grade inflammation in obese individuals by establishing in vitro models similar to the in vivo crosstalk between immune cells/joint cells and adipocytes (which are the major source of cytokines, chemokines and adipokines), and organ-on-chip models have also been proposed [129]. In particular, an in vitro model of lipotoxicity (mediated by palmitate) and cytotoxicity (mediated by inflammatory cytokines) was developed to test the targeting of molecular modulators of sarcopenic Ob [130]. However, to date, these models have not been employed for studying the effect of n-3 PUFAs on ObOA.

**Table 1 ijms-24-09340-t001:** Biological effects of n-3 PUFAs in in vitro models of OA. Up arrows refer to increase and down arrows to decrease.

Molecule	In Vitro Model	Treatment	Main Effects	Specific Outcomes	Ref.
DHA vs. palmitic acid (PA, SFA), oleic acid (OLA, MUFA) and linoleic acid (LA, n-6 PUFAs)	SW1353 chondrosarcoma cells ± LPS (1000 ng/mL)	6.25, 25, 100 μM	Reduced pyroptosis-dependent inflammatory response due to inhibition of TLR4/NF-κB and NLRP3/caspase-1/GSDMD signaling.	DHA, OLA ▪ ↓ TLR4; ▪ ↓ pNF-κB; ▪ ↓ NLRP3, caspase 1, GSDMD; ▪ ↓ IL-1β; ▪ ↑ Coll-II. PA, LA ▪ ↑ TLR4; ▪ ↑ pNF-κB.	[60]
DHA	SW1353 chondrosarcoma ± IL-1β (5–100 ng/mL)	3.125, 6.25, 12.5, 25, 50 µg/mL	Reduced inflammatory-dependent catabolic response by inhibition of p38 MAPK-dependent signaling.	▪ ↓ MMP-13; ▪ ↓ p38 phosphorylation.	[106]
DHA	Human chondrocytes ± TNF-α (50 ng/mL)	25 μM for 48 h	Reduced inflammatory and catabolic response through downregulating Wnt/β-catenin and NF-κB signaling pathways.	▪ ↓ CCL2, COX2, IL-1β; ▪ ↓ MMP-13; ▪ ↓ NF-κB p65 and β-catenin nucleus translocation.	[65]
DHA	Murine chondrocytes ± IL-1β (10 ng/mL)	5, 10, 25 μM, pre-treatment for 24 h	Reduced inflammatory and catabolic response by inhibition of NF-κB p65 and β-catenin by Malat-1.	▪ ↓ nitrites, iNOS, COX2; ▪ ↓ MMP-13; ▪ ↑ Coll II; ▪ ↓ NF-κB p65 subunit nucleus translocation and IκBα phosphorylation; ▪ ↓ Malat 1 (lncRNA).	[111]
EPA/DHA/ALA	Bovine chondrocytes ± IL-1α (10 ng/mL)	2.5, 5, 10, 20, 30 μg/mL for 8 h	Reduced inflammatory and catabolic response, efficiency EPA > DHA > ALA (modulation of more targets, modulation at low concentration).	EPA: ▪ ↓ ADAMTS-4/-5, MMP-3/-13; ▪ ↓ COX-2, IL-1α, IL-1 β, TNF-α; ▪ ↓ COX-1 not affected. DHA: ▪ ↓ ADAMTS-4; ▪ ↓ COX-2, IL-1α, IL-1 β, TNF-α; ▪ ↓ COX-1 not affected. ALA: ▪ ↓ ADAMTS-4/-5; ▪ ↓ COX-2, IL-1α, TNF-α; ▪ ↓ COX-1 not affected.	[112]
EPA/DHA vs. AA	Canine chondrocytes ± IL-1β (10 ng/mL)	10 μM for 8 days	Reduced inflammatory and catabolic response, efficiency EPA > DHA AA positively modulates some markers of inflammation.	EPA: ▪ ↓ iNOS, NO (EPA); ▪ no modulations of MMP-3, 13, ADAMTS4,-5, TIMP-2, COX-2, PGE_2._ DHA: ▪ No modulations of all targets. AA: ▪ ↓ iNOS, NO; ▪ ↑ PGE2, ADAMTS5; ▪ ↓ MMP-3; ▪ no modulations of MMP-13, ADAMTS4, TIMP-2, COX-2.	[120]
EPA/DHA	Bovine cartilage explants ± IL-1β (10 ng/mL)	0.1, 1, 10 μM EPA and/or DHA for 5 days	Reduce cytokine-induced articular cartilage degradation. Efficiency EPA > DHA at long term.	▪ ↓ GAG loss; ▪ ↓ ADAMTS 4-/-5; MMP-3/-13, COX-2.	[113]
Green-lipped mussel (GLM) (abundant in DHA)	Human OA chondrocytes ± IL-1β (20 ng/mL)	10, 100, 250 μg/mL	Reduced inflammatory response and necroptosis.	▪ ↓ RIPK1, RIPK3 and MLKL (necroptotic markers); ▪ ↓ IL-1β, IL-6; ▪ ↓ iNOS; ▪ ↓ MMP-3, -13; ▪ ↓ NF-KB;	[115]
EPA	Normal human knee chondrocytes ± SNP (1 mM)	10, 30, 50 µg/mL for 8 h	Reduced inflammatory-dependent catabolic response and apoptosis by inhibition of MAPK signaling.	▪ ↓phosphorylation of p38 MAPK and p53; ▪ ↓caspase 3 and poly (ADP-ribose) polymerase cleavage; ▪ ↓apoptosis; ▪ ↓ MMP-3, -13.	[114]
RvD1 (DHA metabolite)	Human OA chondrocytes ± IL-1β (1 ng/mL)	0–10 μM	Reduced inflammation by inactivation of NF-κB/p65, p38/MAPK and JNK1/2.	▪ ↓ COX-2, PGE_2;_ ▪ ↓ iNOs, NO; ▪ ↓ MMP-13; ▪ ↓ apoptosis; ▪ ↓ NF-κB/p65; ▪ ↓ p38/MAPK; ▪ ↓ JNK1/2.	[117]
LA (n-6 PUFA) vs. ALA (n-3 PUFA)	Chondrocytes at high density ± IL-1β (100 pg/mL)	LA/ALA (1:1, 2:1, 4:1, 6:1, 8:1, 10:1) total amount 50 μg/mL for 1 h	Anti-catabolic effect, most effective ratio was 1:1, and 10:1 was not effective.	▪ ↓ MMP13.	[121]
PDX	Rat chondrocytes ± IL-1β (10 ng/mL)	Pretreatment, 0.5, 1, 2, 4 μM	Inhibited inflammatory responses through the activation of AMPK and inhibition of NF-κB signaling pathway.	▪ ↑ Coll II; ▪ ↓ MMP-3, -13, ADAMTS4; ▪ ↓ iNOS, COX-2, NO, PGE2; ▪ ↓ NF-κB p65, IκBα phosphorylation; ▪ ↑ IκBα. ▪ ↑ AMPK; ▪ Phosphorylation; ▪ ↓ Nuclear translocation of NF-κB p65.	[118]
Conjugated linoleic acids + AA or EPA linoleic acid (LA) + AA or EPA	Human OA chondrocytes	10 μM	Anti-inflammatory.	▪ ↓ PGE_2_, NO; ▪ LA/EPA: lowest PGE_2_ production; ▪ LA/AA: lowest NO production.	[131]
DHA	Murine bone marrow mesenchymal stromal cells during chondrogenesis ± IL-1β (10 ng/mL)	25 μM, pre-treatment for 21 days	Rescued IL-1β-impaired chondrogenesis by NF-κB signaling inhibition by Malat-1.	▪ ↑ chondrogenesis; ▪ ↑ COLL2, Acan, GAG, Sox-9; ▪ ↓ MMP-13, COLLX; ▪ ↓ NF-κB p65 subunit nucleus translocation and IκBα; ▪ ↓ Malat 1 (lncRNA).	[111]
DHA	293 T cells ± TNFα (10 ng/mL)	5, 10, 25 μM, for 24 h	Downregulated Wnt/β-catenin and NF-κB signaling.	Luciferase activities of reporter vector harbouring Wnt/β-catenin (TOPFlash) and NF-κB response element (NF-κB RE) showed a declining gradient.	[111]
13- and 16-H- DHEA and DHEA (endocannabinoid from DHA)	RAW264.7 macrophage ± LPS (1 µg/mL)	2.5–5 μM	Anti-inflammatory effects less pronounced compared to DHEA.	DHEA ▪ ↓ NO and IL-6; ▪ ↓ TNF-α, IL-1β; ▪ ↓ PGE2, PGD2. 13-H-DHEA ▪ ↓ TNF-α, IL-1β; ▪ ↓ InhbA, Ifit1; ▪ ↑ PPbp, Serpinb2; ▪ ↓ DHA. 16-H-DHEA ▪ ↓ IL-1 Ra; ▪ ↓ InhbA, Ifit1 (downstream to TLR4 activation); ▪ ↑ PPbp, Serpinb2; ▪ ↓ DHA.	[127]
13S,14S-epoxy- DHA (precursor of MaR-1)	Human macrophages (M1 and M2 subsets)	10 nM	Reduced inflammation and switching from M1 to M2 phenotype.	▪ Higher conversion to MaR-1 in M2; ▪ ↓ LTB4 biosynthesis by LTA4; ▪ ↓ arachidonic acid conversion by hm12-LOX; ▪ ↓ CD54, CD80 (M1); ▪ ↑ CD163, CD206 (M2).	[128]
DHA/EPA	Equine synoviocytes ± IL-1β (5 ng/mL)	25–50 μM for 24 h	Reduced inflammatory and catabolic response due to increased integration within cell membranes and production of oxylipids (specialized pro-resolving mediators).	DHA ▪ ↑ Content in the cell membrane; ▪ ↑ RvD1, RvD2, Mar-1, Mar-2, 19,20DiHDPE, 19,20 EpDPE, 17HDoHE, 10,17 DiHDoHE; ▪ ↓ ADAMTS4, MMP-1, -13; IL-1β, IL-6, COX-2; ▪ no significant reduction in TNF-α, MMP-3. EPA ▪ ↑ Content in cell membrane; ▪ ↓ ADAMTS4, MMP-1, -13; IL-1β, IL-6, COX-2.	[108]
RvD-1, -2, MaR-1, PDX	Equine synovial fibroblasts ± IL-1β (5 ng/mL)	Pre-treatment with 25 μM and 50 μM EPA and DHA	Reduced inflammatory and catabolic response due to increased integration within cells membrane.	▪ ↓ ADAMTS4, MMP-1, -13; IL-1β, IL-6, COX-2.	[108]
RvD1	Human OA fibroblast-like synoviocytes (FLs)	20, 50, 100, 200 nM	Reduced inflammatory and catabolic response due to Hippo-YAP signaling pathway activation.	▪ ↓ MMP13; ▪ ↓ IL-1β; ▪ ↑ YAP phosphorylation; ▪ ↓ YAP; ▪ ↓ Proliferation; ▪ (G2 cell cycle arrest; ▪ ↓Cyclin D1, cyclin B1, PCNA; ▪ ↑ p53.	[123]
MaR-1 (DHA metabolite)	Rat FLSs ± IL-1β (10 ng/mL)	Pretreatment, 10, 100, 1000 nM, 1 h	Anti-inflammatory and anti-catabolic effect by stimulation of PI3k/Akt pathway and inhibition of NF-κB p65 pathway.	▪ ↓ MMP13; ▪ ↑ PI3k, Akt phosphorylation; ▪ ↓ pNF-κB p65.	[124]
RvD1/RvE1/MaR1	OA synovial fibroblast ± TNF-α (10 ng/mL)	Pretreatment, 100 nM	No anti-inflammatory effect.	▪ No inhibition of COX-2, mPGES-1, IL-6, MMP-3.	[122]

In vivo preclinical studies could overcome some of the major limitations described for in vitro studies by partially modelling the Ob-induced pathological changes in OA. Notably, a variety of animal models can mimic the varied framework of pathological aspects in OA, including spontaneous (guinea pig model), inflammatory (monoiodoacetate model (MIA)), post-traumatic (anterior cruciate ligament transection (ACLT), destabilization of the medial meniscus (DMM) and medial meniscal transection (MNX)) and pain (MIA and MNX) models (Table 2). Notably, the first indications of the link between OA and tissue alterations of n-3/n-6 PUFAs came from Mustonen AM et al. who showed in a post-traumatic model of OA a decreased n-3/n-6 PUFA ratio in the IFP and an increased pro-inflammatory phenotype [132]. Remarkably, more specific models of ObOA have been generated, integrating classical models of OA and Ob, to survey the connection between OA and Ob and the potential of different molecules in repressing their occurrence/development, including the PUFA family. In this line, different models of Ob have been established and can mainly be classified as (i) genetic (monogenic, polygenic and transgenic models) and (ii) non-genetic (dietary supplementation and surgical (e.g., ovariectomy)) [133,134]. Much attention in experimental studies focusing on ObOA has been paid to (i) the use of HFD and (ii) genetic loss-of-function mutations (e.g., leptin-deficient (ob/ob) and leptin-receptor-deficient (db/db) mice). HFD feeding is one of the most commonly used rodent systems for inducing severe Ob and modelling western-diet-induced Ob [135,136]. An HFD not only increases the intake of triglycerides, SFAs and n-6 PUFAs but activates several biological processes contributing to metabolic dysregulation; it elevates serum glucose and insulin, increases body mass and adipose tissue and induces metabolic endotoxemia (due to increased LPS in circulation) [136]. Notably, a medium-fat diet (MFD), with fat >11%, also induces metabolic changes [118]. An HFD has been shown to aggravate OA synovitis by inducing synovial severe tissue architecture disorganisation, along with the infiltration of macrophages. Preclinical in vivo studies using an HFD regimen in healthy mice showed that it induces OA onset and progression due to increased levels of systemic inflammatory molecules and glucose intolerance [137]; similarly, it displays the accelerated progression of OA in surgical destabilization models of OA. Notably, by employing clodronate-loaded liposomes to deplete local macrophages in the synovial joints, Sun AR et al. demonstrated a clear inhibition of ObOA in HFD-treated mice, thus highlighting the key role of macrophages in eliciting the HFD-induced metabolic effect in ObOA [138]. Recently, the scientific community has mainly focused on the potential contribution of specific FAs. In this respect, Wu et. al. corroborated prior studies showing that n-6 PUFAs play a role as pro-inflammatory agents and are linked to OA, while n-3 PUFAs may act in an anti-inflammatory way and are associated with less OA evolution [15].

The same authors showed that dietary FAs can modulate OA severity differently depending on the content of specific forms of FAs [141]. Besides local effects on articular joints, n-3 PUFAs exert systemic effects by increasing adiponectin expression and reducing leptin and resistin levels, the latter being commonly linked to OA progression. N-3 PUFAs correlate to M2 polarization and adiponectin levels; subsequent further studies have shown that adiponectin in cartilage seems to be indirectly mediated by macrophage polarization towards a wound-healing phenotype [156]

Further studies in several OA models have confirmed that dietary supplementation with several sources of n-3 PUFAs (e.g., DHA, EPA, Antarctic krill oil, triglyceride n-3 oil and PDX) mitigates OA changes (reduction in cartilage degradation, synovial hypertrophy, macrophage infiltration and inflammation) [134,147,148,149,150]. Furthermore, several of these investigations identified the relevant molecular pathways. For instance, GLM demonstrated chondroprotective properties and a decrease in catabolic and inflammatory processes by controlling the necroptotic pathway [115], and low-n-6/n-3 PUFA food oils offset OA progression by inhibiting the NFκB pathway [155]. Notably, 17(R)-HDoHE (RvD2 precursor), aspirin-triggered RvD1 (AT-RvD1), RvE1, RvD1 and Mar2 showed additional analgesic potential in a carrageenan-induced inflammatory model or an MIA model of OA or LPS-induced mechanical hyperalgesia [151,152,153,154]. Altogether, these studies, designed to trace the downstream effects of changing the nourishing properties of animal diets, initially pointed towards a transition in dietary FA composition towards a low ingestion of n-6 PUFAs and SFAs and a high ingestion of n-3 PUFAs to reduce OA.

However, this traditional approach has the following several limitations: high heterogeneity between experimental groups, confounding variables related to the diet and type of FAs chosen, the feeding procedure, the duration of diet administration, impurity or unwanted components and stability issues (e.g., food storage and sensitivity to oxidation), which ultimately lead to conflicting and confusing findings. To overcome these limitations, an innovative first approach was developed for studying the benefits of n-3 PUFAs and their molecular mechanism. To this end, fat-1 transgenic (TG) mice were genetically engineered to carry the fat-1 gene, encoding for the n-3 PUFA desaturase enzyme, which is able to convert n-6 to n-3 PUFAs (which does not exist in mammals) [157]. This model can force the endogenous production of n-3 PUFAs from n-6 PUFAs, thus allowing the study of the beneficial effects of a balanced n-6/n-3 PUFA ratio and an overall increase in n-3 PUFAs in all tissues, thus minimizing the confounding factors of their administration within the diet. Specifically, the fat-1 TG model presents more advantages for studying the benefits of n-3 PUFAs than the HFD model, as it avoids concerns related to the dose, composition and duration of the treatment applied and allows the same diet in both wild-type and transgenic groups [158], thus making the results more reliable. In this way, Kimmerling KA et al. unquestionably demonstrated that (i) FA composition and metabolic inflammation are the primary determinants in the onset and progression of ObOA in a post-traumatic model of OA (DMM model) rather than merely “body weight and mechanical factors” and (ii) that an endogenous reduction in n-6 PUFAs due to their conversion to n-3 PUFAs is beneficial for delaying OA aggravation when both standard diets and HFDs were administered [142]. Interestingly, this group gave evidence of sex differences in the response to n-6 PUFA conversion in favor of females, who displayed lower OA scores [142]. Notably, this preclinical study provided the first evidence of the potential genetic use of n-3 PUFAs desaturase in counteracting post-traumatic OA in obese patients. In line with these findings, Huang et al. showed, in the same TG model, that the endogenous increase in n-3 PUFAs after the administration of a standard diet has chondroprotective potential; moreover, they also gave insights into the molecular mechanism, showing the inhibition of mTORC1 signaling, the promotion of autophagy and cell survival in chondrocytes [143]. Conversely, Cai et. al. showed that the protective effect mediated by increased n-3 PUFAs in fat-1 mice appears to be more pronounced in post-traumatic and inflammatory models of OA than in spontaneous OA [144]. They also found a sex difference in n-3 PUFA conversion; the n-6: n-3 ratio was reduced twelve-fold in males and seven-fold and females [144].

Other models of Ob involve murine genetic mutations with loss-of-function of key genes implicated in ObOA-associated changes. In particular, leptin-deficient mice (ob/ob) and leptin-receptor-deficient mice (db/db), characterized by mutations of the genes encoding leptin or its receptor, cause alterations in leptin-dependent pathways with the subsequent development of severe Ob and diabetes. In particular, ob/ob murine models are widely used as they display the early onset of Ob, and the subjects gain weight in a short period [159]. These models were used to study ObOA models, which develop a more intense degradation of the joint following surgically induced OA. In this regard, Griffin TM et al. underlined the importance of leptin signaling in promoting systemic inflammation, showing that extreme Ob linked to depleted leptin signaling is not sufficient to increase knee OA [145]. A limitation of these (ob/ob) and (db/db) mice model is that they display severe Ob and hyperglycemia that is more representative of a diabetic state than that of metabolic syndrome; furthermore, leptin is involved in mechanisms that affect chondrocyte metabolism and cartilage health, making it difficult to separate the effects of Ob from those of leptin-dependent pathways. To date, only a few studies have been performed with n-3 PUFA treatment on leptin transgenic models. Pinel et al. showed that EPA, and not DHA and ALA, partially induces protection against glucose intolerance and IR, and it increases adiponectin levels and the phosphorylation of Akt in ob/ob mice [146]. Similarly, increased adiponectin levels were found in C57BL/KsJ-db/db mice who were treated for 4 weeks with tetracosahexaenoic acid (THA), DHA and EPA [160].

In summary, molecules that can target inflammation and pain are of valuable interest in the field of OA. N-3 PUFAs, differently from other nutraceutical molecules, hold both features, as documented in several studies [115,139,140,149,151,152,153,161]. Altogether, these findings suggest that dietary n-3 PUFA supplementation could have beneficial effects on ObOA.

## 4. Current and Novel Perspectives of Dietary Interventions in ObOA: Strengths and Weaknesses of the Potential Use of PUFAs

### 4.1. Clinical Evidence of n-3PUFA in OA: Insights for Better ObOA Therapies

Numerous clinical studies have demonstrated the link between Ob and OA, wherein changes in joint movement and systemic factors, such as impairment in the gut microbiome (gut dysbiosis) and of circulating adipokines and FAs, have a significant effect on joint metabolism and lead to a high level of inflammation in ObOA patients [15,87,91,162,163].

Therefore, identifying strategies that can reduce weight, target dysbiosis and switch off the inflammatory state to restore compromised joint metabolism is very challenging in ObOA patients. Among the various desirable perspectives in this field, exercise control and the reformulation of the dietary regimen may represent two alternative adjuvant treatments. Currently, 34 clinical trials on ObOA have dietary control as a therapeutic strategy (https://clinicaltrials.gov/ct2/results?cond=osteoarthritis+and+obesity&term=Diet+&cntry=&state=&city=&dist= (accessed on 13 May 2023)), providing evidence for the importance of dietary interventions to improve physical function and reduce pain in patients with ObOA. Interestingly, these studies also gave evidence for the role of prebiotics in modulating the health of microbial communities towards a protective phenotype, reducing the severity of inflammation. Concerning nutrition-based strategies exploring FAs, they are currently largely being studied in the obese cohort, mainly due to the preclinical evidence for the anti-inflammatory and anti-nociceptive properties of n-3 PUFAs (see Table 1 and Table 2). Currently, 123 clinical trials on Ob have involved the administration of FAs, targeting inflammation as one of the major outcomes (https://clinicaltrials.gov/ct2/results?cond=obesity+and+fatty+acid&term=&cntry=&state=&city=&dist= (accessed on 13 May 2023)). However, to date, no active clinical trials involve the use of n-3PUFAs in ObOA as a dietary administration to prevent OA development and inflammation. Only a randomized, double-blind, placebo-controlled trial of 16 weeks was performed on fish oil supplementation and demonstrated the beneficial effect on OA-related pain in older adults with Ob/overweight [164].

In general, there is a growing consensus in the scientific community on the health benefits of n-3 PUFAs in various inflammatory disorders, including OA [58,121]. Conversely, a high-n-6-PUFA diet has been highly associated with OA-associated symptoms (synovitis) [165]. Given these premises, several clinical studies have evaluated the effects of disparate sources of EPA and DHA, n-3 PUFA compounds, either independently or in combination with other drugs (e.g., aspirin) or other dietary supplements (e.g., glucosamine) to counteract OA progression (Table 3).

The first study that investigated the role of n-3 PUFA in OA patients dates back to 1989 [166]. Participants in this study underwent oral supplementation with cod liver oil containing EPA, together with anti-inflammatory treatment with ibuprofen, for six months, experiencing no beneficial effects. In a longer follow-up study lasting 24 weeks, the clinical results of patients taking cod liver oil showed no benefit from the treatment, confirming previous short-term results [167]. Further clinical studies, however, have highlighted the biological relevance and clinical significance of EPA and DHA in protecting joint tissues. The use of GLM, an extract rich in EPA and DHA, has been shown to improve joint stiffness and reduce analgesic consumption compared to a placebo group and is more tolerated as an oral supplement by patients [168]. In line with this study, Lau CS and his group demonstrated that GLM exerts anti-nociceptive effects already after 8 weeks of treatment and improves several OA assessment parameters [169]. These positive clinical results of GLM supplementation have been attributed to its content of EPA and DHA, which exert anti-inflammatory effects by inhibiting the COX and LOX cascades of AA metabolism, with a subsequent reduction in PGs and leukotrienes [170]. In addition, scientific indications have reported further protective effects from the presence of other pre-resolving lipid mediators within the composition of GLM [171]. In line with these findings, Coulson et al. demonstrated, in a clinical trial conducted on OA patients, the impact of GLM extract or glucosamine in restoring the gut microbiome by reducing the Clostridia and Staphylococcus genera and improving joint pain, function and stiffness [172]. Besides their prebiotic effects, both GLM extract and glucosamine reduced inflammatory processes, as a reduction in Clostridia is a potent modulator of Th17 and CD4^+^ regulatory cells.

Further proof of the advantages of n-3 PUFAs was obtained from a study that monitored the effects of a commercial dietary supplement containing fish oil with omega-3- and omega-6-rich foods, *Urtica dioica*, zinc and vitamin E for 3 months. In particular, patients showed a significant reduction in pain, improved stiffness and function and used fewer analgesics than the control group [173]. Dietary supplementation with PUFAs reduced patient-reported joint pain intensity, morning stiffness duration, painful and/or tender joints and the use of non-steroidal anti-inflammatory drugs (NSAIDs); the results of another meta-analysis showed a reduction in NSAID consumption with the use of n-3 PUFAs.

Taken together, these studies provided clinical evidence of the protective role of n-3 PUFAs in OA, although they did not provide scientific evidence on the specific dose of each dietary component to obtain optimal doses of n-3 PUFAs. To fill this knowledge gap, Hill C.L. et al. compared the clinical outcomes of 202 OA patients treated for two years with a low (low-dose fish oil and high-dose oleic oil at a ratio of 1:9, resulting in 0.45 g EPA + DHA) or high (18% EPA and 12% DHA using 4.5 g EPA + DHA) dose of n-3 PUFAs, respectively. Interestingly, the low-dose oral n-3 PUFAs provided better benefits than the high-dose ones in reducing pain and improving joint function [174]. Although this study provided initial evidence for the dose-dependent effects of n-3 PUFAs, further studies are necessary to confirm these results. Another oil rich in EPA and DHA that has attracted increasing interest is krill oil, which is extracted from zooplankton. The first clinical evidence for krill oil’s role in counteracting chronic inflammation in OA dates to 2005 with a randomized, double-blind, placebo-controlled study. Oral supplementation with krill oil doses of 300 mg/day and 2 g/day was effective in reducing serum inflammation and improving joint function with a significant decrease in pain in two randomized controlled clinical trials [175,176]. However, these RCT studies showed some limitations as they only considered a short duration (30 days), without considering potential adverse effects at a longer follow-up. Therefore, scientists have conducted further studies on krill oil to assess its long-term effects. These clinical studies provided insights into the safety profile of krill oil and confirmed its protective role with oral supplementation in relieving pain in OA patients and restoring stiffness and physical function in patients [177,178]. In line with these findings, a multicenter, randomized, double-blind, placebo-controlled trial of 235 OA patients taking oral supplementations of 4 mg/d of krill (Euphausia superba) oil, which is abundant in EPA and DHA, provided firm evidence for the improvement in joint pain, stiffness and physical function in patients with mild-to-moderate knee OA [178]. Further confirmation of the specific benefits of EPA and DHA also emerged from a study in which they were administered as oral supplements as adjuvants to glucosamine sulphate, a drug used to treat OA [179]. They were effective in reducing pain and OA symptoms. In another clinical study, EPA was administrated along with l-serine (l-Ser), an amino acid that is essential for maintaining the normal function of the nervous system, improving pain score in healthy subjects [180]. Considering the protective effects observed for resolvins, some clinical studies have focused on evaluating the role of RevD1, RevD2 and 17-HDHA, DHA in heat pain sensitivity and OA pain in humans by liquid chromatography–mass spectrometry. The findings from this clinical trial highlighted, for the first time, the potential of 17-HDHA as a biomarker for pain [181]. Currently, a clinical trial is carrying out the administration of PF-04457845, an inhibitor of fatty acid amide hydrolase (FAAH), a serine hydrolase with a prominent role in the hydrolysis of endocannabinoids, and metabolites of n-3 PUFAs for reducing chronic pain (NCT00981357); no data are currently available.

Overall, the clinical studies performed on OA patients have provided evidence-based findings on the health benefits of n-3 PUFAs in OA. Given the recent interest shown by the scientific community in studying FA administration to control Ob-induced pathological changes (123 open clinical trials) and dietary control as a therapeutic strategy in ObOA, we believe that these clinical studies of n-3 PUFA administration in OA patients lay the ground for forthcoming studies on the ObOA cohort.

**Table 3 ijms-24-09340-t003:** Human studies and clinical trials of n-3 PUFAs in OA patients. Up arrows refer to increase and down arrows to decrease.

Molecule Tested	Study Types	Patient Data	Treatments/ Follow-Up (F.up)	Main Effects	Ref.
FA intake	Prospective study.	N = 2092 participants with radiographic knee OA.	Followed at yearly intervals up to 48 months. Questionnaire for food intake.	Significant positive relationships between total fat and SFA with joint space width loss were observed. MUFA, PUFA and a higher ratio of PUFA to SFA were associated with a reduced joint space width loss.	[182]
Fasting plasma phospholipid n-6 (AA) and n-3 PUFAs (EPA and DHA) with synovitis	Multicenter Osteoarthritis Study (MOST).	N = 472 patients with knee OA (50% women). Mean age = 60 year. BMI = 30 (1° grade of Ob).	n-3 PUFAs. n-6 PUFAs.	Multivariable logistic regression showed the following: ▪ Positive effects of plasma levels of n-3 PUFAs with patellofemoral cartilage; ▪ Negative effects of n-6 PUFAs in mediating synovitis.	[165]
Fish oil (FO) (DHA + EPA)	Randomized, double-blind clinical study.	N = 152 older adults between 50 and 80 years. BMI 25–40 kg/m^2^	*Group 1*—*FO* 2000 mg/day DHA + 400 mg/day EPA). *Group 2*—*CUR* curcumin (160 mg/day). *Group 3—FO + CUR.*	▪ FO reduced OA-specific pain and burden; ▪ FO improved microvascular functions; ▪ CUR alone and in combination with FO did not reduce pain measures.	[164]
Cod liver oil (EPA)	A double-blind, placebo- RCT.	N = 26; Female, *n* = 21; Age range = 52–85 years.	*Group 1*—EPA oil (10 mL/d EPA) and ibuprofen (1200 mg/d). *Group 2*—placebo (oil of undescribed content) and ibuprofen. F.up = 6 months.	▪ No significant differences in pain and daily activities.	[166]
Cod liver oil (EPA + DHA) + NSAIDs	A double-blind, placebo- RCT.	N = 86; female, *n* = 60; Age range = 49–87 years.	*Group 1*—cod liver oil (10 mL of oil containing 786 mg EPA) + NSAIDs. *Group 2*—Placebo (10 mL olive oil) + NSAIDs. F.up = 24 weeks.	▪ No significant benefit was observed for patients taking cod liver oil compared with the placebo group.	[167]
GLM (high proportion of EPA and DHA + low presence of several minor lipid components)	A double-blind, placebo- RCT.	N = 80; Female, *n* = 44; age = 66.4 ± 10 years. Pain rated > 30 mm in the last week on 100 mm VAS.	*Group 1*—GLM extract (600 mg/d). *Group 2*—Placebo (600 mg/d corn oil). F.-up = 0, 6, 12 and 15 wks.	▪ No pain improvement; ▪ Beneficial effects on stiffness; ▪ Reduction in acetaminophen treatment in the post-intervention period.	[168]
Lyprinol^®^ (a lipid extract of GLM rich in EPA and DHA)	A double-blind, placebo- RCT.	N = 80 patients with knee OA female, *n* = 69 knee pain, radiographic evidence of osteophytes.	GLM group—four capsules of Lyprinol^®^/day. Placebo group—olive oil in the same number of capsules. F.up = 6 months. Revision at week 0, 2, 4, 8, 12, 18 and 24.	▪ Safe and well tolerated; ▪ ↓ Knee pain in the GLM group; ▪ Improvement in the patient’s global assessment of arthritis.	[169]
GLM extract	Non-blinded randomized clinical trial.	N = 38 patients with knee OA.	*Group 1*—GLM extract (3000 mg/day). *Group 2*—glucosamine (3000 mg/day). Treatment for 12 weeks.	▪ ↓ Clostridium and Staphylococcus species in both groups; ▪ ↑ Lactobacillus, Streptococcus and Eubacterium species; ▪ Improvement in the GI tract; ▪ ↓ Joint pain.	[172]
Phytalgic^®^ (fish oil rich in n-3 PUFA+ n-6 PUFAs+ vitamin E, *Urtica dioica*)	Randomized double-blind parallel-groups clinical trial.	N = 81 patients with OA of the knee or hip using NSAIDs and/or analgesics regularly. Female, *n* = 55; Mean age = 57.5; Age range = 28–84 years) F.up = 3 months.	*Group 1*—Phytalgic^®^ (*n* = 41). *Group 2*—placebo (*n* = 40).	▪ ↓ WOMAC score for pain stiffness; ▪ ↓NSAID use.	[173]
EPA+ DHA	A randomized, double-blind, multicenter trial. Trial registration number ACTRN 12607000415404.	N = 202 patients with knee OA. Female, *n* = 100; Mean age = 61 ± 10 years, Participants were >40 years with clinical knee and VAS > 20 mm, No indication of BMI.	*Group 1*—hHigh-dose fish oil (4.5 g EPA + DHA per day) (59% women) 15 mL/day. *Group 2*—low-dose fish oil (0.45 g EPA + DHA per day) (40% women).	▪ The low dose had a greater improvement in WOMAC pain and function scores at 2 years; ▪ No difference between the two groups in cartilage volume loss.	[174]
Neptune Krill Oil (NKO^TM^) EPA (20:5 n-3) + DHA (22:6 n-3) + antioxidants (e.g., astaxanthin, etc.)	A randomized, double-blind, placebo-controlled study.	N = 90 patients with cardiovascular disease and/or rheumatoid arthritis and/or OA and high levels of CRP (>1.0 mg/dL).	*Group 1*—treatment with NKO™ (300 mg daily). *Group 2*—placebo 30 days of treatment.	▪ ↓ CRP; ▪ ↓ WOMAC pain score; ▪ ↓ Pain; ▪ ↓ Functional impairment.	[175]
Krill oil (EPA (20:5 n-3) + DHA (22:6 n-3))	Randomized, double-blind, parallel-group, placebo-controlled trial.	N = 50 patients with mild knee pain (no severe pain).	*Group 1*—treatment with 2 g/day. *Group 2*—placebo. F.up = 30 days.	▪ ↓ Knee pain; ▪ ↑ Plasma EPA and EPA/AA ratio.	[176]
Krill oil (EPA (20:5 n-3) + DHA (22:6 n-3))	Multicenter, randomized, double-blind, placebo-controlled clinical trial.	N = 260 patients with clinical knee OA, significant knee pain and effusion-synovitis.	*Group 1*—treatment of 2 g/day. *Group 2*—placebo. F.up = 6 months.	▪ Safe ▪ ↓ Knee pain ▪ Reduced sizes of knee-effusion synovitis.	[177]
Krill oil (EPA (20:5 n-3) + DHA (22:6 n-3))]	Multicenter, randomized, double-blind, placebo-controlled trial	N = 235; Female, *n* = 129; Mean age = 55.9 ± 6.8 yrs; BMI > 18.5 to <35 kg/m^2^; Mild-to-moderate knee OA.	*Group 1*—4 g/d of a commercially available krill oil supplement daily (0.60 g EPA/d, 0.28 DHA/d, 0.45 mg astaxanthin/d). *Group 2*—placebo (4 g/d mixed vegetable oil). F.up = 6 months.	▪ ↓ Knee pain ▪ Improved stiffness and physical functions; ▪ No changes in NSAID use.	[178]
Combination of glucosamine sulfate + EPA DHA	RCT, a double-blind study.	N = 177 patients with moderate-to-severe hip or knee OA. Mean age 62 y; mean BMI = 29; 63% women.	*Group 1*—glucosamine sulfate + EPA DHA. *Group 2*—glucosamine sulfate alone.	▪ ↓ WOMAC pain score superior in group 1; ▪ ↓ OA symptoms (morning stiffness, pain in hips and knees) were superior in group 1.	[179]
EPA + l-serine	Randomized, double-blind, placebo-controlled, parallel-group study.	N = 120 participants aged ≥ 20 y (36 men and 84 women: mean ± SD age = 40.8 ± 10.9 year.	*Group 1*—oral administration of 549 mg l-serine+ 149 mg/daily EPA. *Group 2*—placebo group. 8 wk dosing and 4 wk post-treatment observation.	▪ ↓ Pain score.	[180]
Resolvins D1, D2 and 17-HDHA, DHA	OA case-control cohort.	N = 62 individuals affected with radiographic knee OA (Kellgren–Lawrence grade of 2 or higher). 52 individuals without radiographic or clinical symptoms of OA.	No treatment gas chromatography	▪ 17-HDHA was associated with pain: ↑ thermal pain sensitivity and intensity of chronic pain; ▪ No evidence of pain-related features for other studied resolvins.	[181]

### 4.2. Nutritional Recommendations for ObOA Patients

There is mounting evidence that clinical decision-making in OA should be reinterpreted in the context of an evolving Ob care model with a focus on establishing nutritional and metabolic clinical practice guidelines [183]. In overweight/obese patients, nutritional recommendations are of paramount importance in foremost promoting weight reduction and decreasing the load on weight-bearing joints. Indeed, weight loss can improve both pain and function in obese patients with OA, regardless of the severity of the joint [184]. Moreover, the introduction of balanced nutrition and a healthy diet can modulate several mechanisms related to ObOA, including inflammation, dysbiosis and immune regulation, thereby improving patients’ “quality of life” and decreasing healthcare costs [162].

Evidence-based recommendations are also mandatory to educate patients on how to manage OA symptoms while preserving their health, as a high percentage of patients with OA take various forms of dietary supplements (even without scientific recommendations) to improve their condition [185]. Several preclinical studies have indicated the efficacy of specific nutrients, vitamins, antioxidants and other natural compounds considered part of a normal diet in cartilage metabolism and their involvement in knee OA by reducing inflammation, promoting antioxidant pathways and increasing weight loss [186,187]. However, rigorous clinical studies that assess the effectiveness of these compounds are still missing due to the low number of trials and high heterogeneity in study design and protocols [188,189]. Moreover, the heterogeneity of the existing literature and the paucity of randomized, controlled clinical studies in humans makes it difficult to design adequate amounts of dietary intake that are able to reach therapeutically relevant concentrations in tissue to exert anti-inflammatory actions [19]. Ob, a known strong risk factor for OA, is often accompanied by dysregulated lipid metabolism with elevated levels of FAs in blood, SF and tissues and an altered ratio between n-3 and n-6 PUFAs [190]. Several studies have confirmed that the type and quality of fat consumed in the diet have a huge impact on the risk of developing OA. In particular, a western diet (characterized by high-fat dairy products, refined grains, a high consumption of red and processed meat and sugar-sweetened beverages and low consumption of fresh fruits, vegetables and legumes) contains higher levels of SFA and n-6 PUFAs than n-3 PUFAs at a ratio of (15–30):1, which predisposes consumers of these diets to inflammation [191] and the onset of several chronic inflammatory processes (among which are Ob and OA) [192]. Similarly, a study found that 75% of traditional Middle Eastern dishes were low in PUFAs, and in 96% of them, the polyunsaturated-to-monounsaturated ratio was below the recommended ratio of 1:1:1 [193]. Notably, it has been clarified that the ratio of n-3 PUFAs to n-6 PUFAs, which should be ideally (4–5):1 [187,192] and is extremely important for promoting beneficial biological effects. Dietary recommendations on dietary FAs have been proposed for other chronic pathologies such as coronary heart disease [194]. Overall, the intake of FAs in general populations worldwide does not meet dietary recommendations, although some virtuous dietary patterns exist. Two examples include the Mediterranean diet, which is characterized by copious amounts of fruits, vegetables, legumes, whole grains, nuts and seafood and the use of olive oil as the main fat source [83], and the Japanese diet, which relies on a diminished consumption of red meat, milk and dairy products and a larger consumption of fish and seafood, resulting in a lower consumption of SFAs and a higher consumption of n-3 PUFAs. Therefore, it is essential to decrease the omega-6 intake while increasing the omega-3 intake in the prevention and management of ObOA [192] and to shift the balance of eicosanoids toward a more beneficial direction, as well as to modulate downstream cannabinoid compounds, which can represent useful biological targets to provide pain relief, especially in OA patients with chronic pain. Similarly, replacing SFAs with MUFAs activates beneficial anti-inflammatory mechanisms (M2 macrophage polarization, the secretion of IL-10 by adipocytes and the inhibition of NLRP3 inflammasome) and reverses the deleterious effect of SFAs on adipose tissues.

Another important piece of dietary advice is to prefer supplementation with EPA and DHA as opposed to ALA. ALA is often the main dietary n-3 PUFA; however, it is rather poorly converted to EPA and DHA [195]. Indeed, although mammals have the essential enzymes used in this pathway, it has been estimated that only about 5% of ALA is converted to EPA, while less than 0.5% is converted to DHA in humans [196]. In addition, a low content of n-3 PUFAs in blood and peripheral tissues is also caused by the competition of n-6 PUFAs LA for elongation and desaturation in the same enzymatic pathway to produce AA, thus substantiating again the importance of reducing dietary intake of n-6 PUFAs. Besides correcting dietary fat content, the worldwide increase in Ob is also related to the lipogenic capacity of certain foods, dietary habits and metabolic states of insulin resistance, all of which contribute to lipogenesis and Ob. In particular, a high carbohydrate content is a key risk factor for high blood sugar, which can be converted with the help of insulin into even-chain SFAs and subsequently into MUFAs through the process of de novo lipogenesis. These FAs are then delivered to fat tissues for storage [197]. Along this line, a randomized, double-blind, parallel-controlled trial on type II diabetes (T2D), another comorbidity factor contributing to OA, showed that the supplementation of a low-carbohydrate, high-protein (LCHP) diet with an n-3 PUFA-rich diet helps to control T2D and is therefore indirectly beneficial for OA [198]. Regarding proteins, there is a divergence in opinions concerning the relationship between dietary protein and inflammation, and this might depend on the protein source and amino acid composition. In particular, protein-rich foods of animal origin (e.g., red meat, eggs and dairy products) often come with a high concentration of saturated fat and cholesterol. There is the potential that consumers of high-protein diets could be exposed to a heightened risk of heart disease, hyperlipidaemia and hypercholesterolaemia. Conversely, plant-sourced proteins (soy protein, beans, tofu, seitan and nuts) or proteins from seafood could be a suitable option. Additionally, any extra protein is converted into glucose (via gluconeogenesis) or ketone bodies. Therefore, favouring a high protein content with plant sources may have considerable benefits in terms of satiety and weight control for obese individuals, avoiding the side effects related to the high fat content of animal proteins [199]. Interestingly, recent studies have shown that the fermentation of dietary fiber by colonic microbes generates short-chain fatty acids (SCFAs), lower levels of which are associated with increased inflammation and metabolic syndrome. Dietary interventions with a high fiber intake may therefore be effective in reducing visceral fat mass, and observational data indicate that this may also be beneficial in alleviating OA pain [200].

Supplementing mice diets with oligofructose fiber may reverse Ob-related gut microbial changes and protect mice from developing OA [86]. Overall, reformulating dietary supplementation by controlling lipid mediators, carbohydrates and protein in a balanced ratio seems to be promising among other therapeutic strategies, especially in ObOA, where it could also help to restore the correct ratio of FAs and restore bacterial diversity in the gut microbiota, which may predispose OA onset and progression [201], e.g., by downregulating the ratio of *Firmicutes/Bacteroidetes* that have been shown to be increased in the gut microbiota in both preclinical models [202] and obese patients [203].

However, to boost innovation in this field and reduce the time needed to reach a translational application, there is an urgent need to act on many aspects at various levels of society to (i) optimize clinical studies for a better interpretation of clinical results and (ii) drive societal changes.

First, the community and patients should be sensitized to the importance of participating in clinical trials, as society would benefit from their participation. Moreover, efforts should be directed to limit one of the major problems associated with clinical trials on dietary molecules, namely dietary adherence. A greater awareness, both in patients and in the community, of the importance of maintaining strong adherence to the proposed diet regimen would help in obtaining more robust clinical results and avoiding confounding results. The second critical aspect is related to the large variability in the response to n-3 PUFAs due to intra-individual differences in OA phenotypes related to the metabolic state, gut microbiota composition and genetic profile [204,205,206]. Clinicians should design clinical trials by stratifying patients according to their genomic, proteomic and lipidomic profiles in order to reduce the variability in responses. In particular, gender differences in lipid metabolism should be considered, as there is great variability in lipid metabolism between men and women (e.g., the conversion of ALA to EPA and DHA is more efficient in women than in men) [207]. However, the high cost of some types of analysis coupled with the high level of specialization of experts and the scarcity of instruments contribute to severely limiting the possibility of stratifying the population. Moreover, clinical studies should be encouraged to report on whether some patients do not respond to “n-3 PUFA treatment” in order to lay the ground for the identification of potential factors beyond responder and non-responder patients and lead a better patient stratification. Another critical aspect is the physicians’ choice of the food molecule to be studied, considering individual preferences and safety. Proposing dietary strategies based on fish would exclude vegans and vegetarians, for whom chia seeds, flaxseed, camelina, watercress or seed oil in low-heat cooking [208] are the only alternatives to fish for the intake of ALA, EPA and DHA. Similarly, including fish-derived molecules in clinical trials could be discouraged (or a reformulation of doses and frequencies of use considered) given global pollution, which leads to potential alterations in the content and quality of n-3 PUFAs in food because of contaminants. Indeed, the bioaccumulation of harmful substances (mercury, microplastics, etc.) in wild fish is a serious problem, as they accumulate in the adipose tissue of these species that supply LC-PUFAs [209]. High-risk populations, such as children and pregnant women, must receive more attention. In this regard, Storelli M.M. proposed limiting the consumption of these foods in such groups, replacing fish-based meals with the use of vegetable oils [210]. The search for food alternatives should be encouraged, and clinical evidence and safety profiles should be provided. In this light, krill oil being extracted from zooplankton, which is at the bottom of the food chain, makes it an ideal choice since it is not polluted with heavy metals, dioxins and pesticides. Another important aspect to consider is that the various dietary habits in different countries around the world (Mediterranean diet, African diet, etc.) of subjects enrolled in clinical trials may complicate the interpretation of the results. Finally, diet as opposed to oral supplements could generate variability due to more uncontrolled bioavailability.

The combination of all these aspects, involving distinct figures and systems, may help to reduce the variability observed in studies and the risk of confounding results in clinics (Figure 4).

Similar aspects should be considered to drive societal changes. Patients and the entire community should realize the importance of a balanced diet in achieving health benefits both for therapy and prevention and the potentially harmful role of saturated lipids and an incorrect ratio of n-3/n-6 PUFAs. Various policy initiatives should raise awareness of the protective role of a good nutritional strategy. In addition, information initiatives dedicated to vegans and vegetarians should inform about supplementation alternatives with plant sources of PUFAs, since they do not eat fish and thus may be predisposed to lower levels of EPA and DHA intake than meat-eaters [211]. Concerted action by the Ministry of Health and the Government to raise awareness of the importance of balanced nutrition and its impact on health is important. To this end, there are an increasing number of awareness-raising campaigns that have been launched on various communication platforms to motivate and encourage adherence to therapeutic nutritional regimes applied in studies [196,212]. In this context, further policy measures should be implemented to offer new treatment paths in the future, which are more and more being aimed at ensuring patient health and shortening the recovery path. Finally, at a global level, it is crucial to promote societal change in terms of environmental policy by putting the serious safety risk of fish pollution at the center of the political discussion.

### 4.3. PUFA-Based Approaches for OA: From Current Evidence to New Perspectives for Intra-Articular Administration

Developing new therapeutic strategies to administer n-3PUFAs in knee joints is a major challenge for clinicians, who must overcome several limitations that are mainly linked to safety and stability issues (Figure 5).

First, the search for solutions to counteract the generation of toxic products is required. Second, it is difficult to design adequate amounts of dietary intake of foods rich in PUFAs that are able to reach therapeutic concentrations in tissue to exert anti-inflammatory and anti-nociceptive actions. Several approaches have been developed to improve bioavailability, including esterification, e.g., in triglycerides (TGs), ethyl esters (EEs) or monoglycerides (MAGs) [213]. However, there are controversial results on what esterification form provides superior bioavailability due to differences in study designs and dietary intakes.

Scientists have attempted different ways to improve the oxidative stability of PUFAs by (i) the addition of natural plant-derived extracts and/or synthetic antioxidants or metal-chelating agents [201,214] and (ii) the use of encapsulation methods (Table 4). In general, it has been demonstrated that marine oils, such as Antarctic krill (*Ephausia Superba*, a zooplankton crustacean) oil, which contain EPA and DHA in addition to antioxidants, provide greater bioavailability than fish oils [215,216,217]. Among antioxidants, synthetic antioxidants include DL-α-tocopherol, butylated and ascorbyl palmitate (AP). Natural products such as polyphenols, carotenoids and tocopherols have gained particular attention due to their beneficial effects [201,218].

In general, encapsulation prevents direct contact between PUFAs with ROS and inflammatory mediators, with important insights in the OA context characterized by an inflammatory framework. There are different strategies for protecting and encapsulating oils rich in PUFAs. First, lipid nanoparticles are largely employed to protect and deliver hydrophobic bioactive compounds. In this regard, nanoemulsions, nanoliposomes, nanoparticles and lipid nanocapsules have been mainly proposed. Hydrogels, oleo gels, nanocrystals and nanofibers represent other carriers used to encapsulate PUFA oil [219].

Besides its safety role, nanoencapsulation ensures more chemical stability and high bioavailability in the GI tract and thus an increased biological activity in damaged joint tissues and an improved efficacy with low side effects (reduced formation of toxic bioproducts), and it holds the possibility of controlled and targeted release [220] while improving biological effects [208,219,221,222]. Similarly, microencapsulation (the transformation of fluid fish oil into a powdered form) provides significant improvements in its handling, storage and stability [223]. To date, several vegan n−3 PUFA products synthesized via the microencapsulation technique are commercially available with good nutritional content [208]. Currently, PUFAs that have been effectively encapsulated include α-linolenic acid, DHA, LA, polyunsaturated lipids (fish oil) and Mar-2 [224,225,226,227,228,229,230,231,232,233].

Taking into account the restrictions that limit the utilization of oral PUFAs and the systemic side-effects on the GI tract of some formulations, the opportunity of encapsulating PUFAs poses the basis for improving IA-based delivery strategies.

The above-mentioned technological approaches are now being increasingly used in the context of preclinical in vivo studies for overcoming bioavailability issues and assessing the efficacy of n-3 PUFAs in ObOA models and inflammatory OA models.

RvD1 is a small molecule that requires frequent administration because its action is rapidly lost in vivo due to lymphatic clearance or oxidation-mediated deactivation. To overcome the problems related to bioavailability, an approach considering direct IA administration has been pursued, and it has been demonstrated that RvD1 has a great anti-inflammatory effect in ObOA that is also mediated by direct action on the native immune system by reducing M1 macrophages [138]. Other studies have considered different approaches to further protect RvD1 from degradation. Dravid A et al. demonstrated that Lipo-RvD1, an RvD1-loaded nano-liposomal formulation, showed a long-lasting effect in a preclinical in vivo model of OA, particularly promoting the resolution of OA-associated inflammation by promoting M2 macrophage polarization [222]. Recently, the same group showed that the encapsulation of RvD1 in liposomes drives the protective effects and sustained release of RvD1 liposomes in the treatment of ObOA [234]. In particular, an increased number of pre-resolution M2 macrophages and a reduction in M1 in obese mice contributed to reducing the inflammatory process and restoring joint homeostasis [234]. Interestingly, benefits were also observed following the IA injection of Mar-1 (two treatments/week for 4 weeks), which halted OA progression in an MIA-induced inflammatory model of OA by acting on both cartilage and synovium [124]. Similarly, Tsubosaka et al. demonstrated that degradable gelatin hydrogels containing EPA may be more effective than a single IA injection of EPA in preventing OA progression in a DMM-induced post-traumatic OA rat model, ensuring a gradual release within 3 weeks [235].

Along this line of research, Soliman M. et al. proposed another nanotechnological approach to enhance the chemical and physical stability (in the circulation) of date seed oil (DSO) in a carrageenan-induced rat paw oedema to reduce inflammation [236]. In particular, this group proposed the use of niosomes with a size range from 100 to 200 nanometres, which were capable of transporting amphiphilic, lipophilic and hydrophilic molecules, thus ensuring efficiently targeted medication in the knee joint [237]. Although niosomes have similar structures to liposomes, they use nonionic surfactants, thus ensuring a higher stability and sustained release properties [238]. There are several noisome-based applications to treat OA with promising results for promoting an anti-inflammatory and analgesic effects [239]. Niosome application in DSO induced a sustained release pattern and anti-inflammatory action [239].

Notably, a microemulsion-incorporated dissolving microneedle co-loading celecoxib and α-linolenic acid (Cel-MEs@MNs) was designed to overcome the incapability of directly loading hydrophobic components by dissolving microneedles, which limits the application of transdermal drug delivery [240]. This study is of remarkable importance, as it demonstrated low swelling, reduced inflammatory cytokines and improved cartilage repair in an OA model compared with oral administration [240].

Although this field is still in its infancy and involves the combination of a few molecules, there are good prospects to continue in this direction to ensure the safety and optimize the bioavailability of these n-3 PUFAs in knee joints in ObOA models.

**Table 4 ijms-24-09340-t004:** Preclinical in vivo models studying models to increase n-3 PUFA bioavailability or intra-articular applications. Up arrows refer to increase and down arrows to decrease.

Molecule	In Vivo Model	Dose/Delivery Route	Main Effects	Specific Outcomes	Indications for Preventive-Therapeutic Strategies for ObOA	Ref.
RvD1	ObOA model: DMM model C57Bl/J6 mice + HFD (45 kcal% fat) vs. control diet (10 kcal% fat).	20 ng/μL (one week before and on weeks 1 and 6 after OA induction). IA injection	Protective role of IA injection of the pro-resolving RvD1 in modulating macrophage phenotype to counteract inflammation.	▪ ↓Cell density and thickness in the synovium; ▪ ↓ IL-1β, Cxcl10, TNF, IL6 and Ccr7 (M1 markers); ▪ ↓F4/80^+^CD11b^+^MHC II high; ▪ ↓ COL10, DIPEN, NITEGE; ▪ ↓ M1 in synovium; ▪ ↑ M2 polarization.	The potential of targeting macrophage phenotypes to prevent OA aggravation.	[161]
RvD1-loaded nano-liposomal formulation (Lipo-RvD1)	Post-traumatic OA model: DMM model in male C57BL/6 mice.	Liposomes (~1 mg per joint in a total volume of 10 μL). IA injection.	Lipo-RvD1 formulation could be a therapeutic candidate thanks to its anti-inflammatory and analgesic properties.	▪ Liposomes increased the IA retention of RvD1; ▪ ↓ M1 macrophage; ▪ ↑ M2 macrophage; ▪ ↓ Osteophyte formation; ▪ ↓ Pain.	The potential of targeting macrophage phenotypes to prevent OA aggravation with analgesic effects.	[222]
RvD1	ObOA model: DMM model + HFD.	RvD1 encapsulated in liposomes (lipo-RvD1).	Improved joint health following the treatment with the lipo- than the free RvD1 treatment.	▪ ↓ Cartilage degradation; ▪ ↑ M2/M1 ratio in synovium.	The potential of lipo-RvD1 as an anti-OA agent.	[234]
MaR-1	Inflammatory model of OA: MIA model in Sprague–Dawley rats.	10 ng MaR-1 + 50 µL sterile saline (two treatments per week for 4 weeks). IA injection.	Chondroprotective effects in mitigating OA progression.	▪ ↑ Coll II; ▪ ↓ MMP13 in the synovium.	The potential of promoting cartilage repair.	[124]
Gelatin hydrogels with EPA	Post-traumatic OA: DMM model in mice.	*Group 1*—SHAM. *Group 2*—DMM. *Group 3*—DMM + corn oil. *Group 4*—EPA-I (DMM + corn oil and EPA). *Group 5*—control (DMM + gelatin hydrogels). Group 6—EPA-G (DMM + gelatin hydrogels containing EPA). IA injection.	Hydrogel incorporating EPA was more effective in attenuating the inflammatory effects underlying the progression of OA.	Gelatin hydrogels containing EPA were more potent compared with a single EPA injection through: ▪ ↓ M1 macrophage; ▪ ↓ CD86^+^ cells; ▪ ↓ F4/F80; ▪ ↓ IL-1β, p-IKK, MMP-13; ▪ Gradual release of EPA (average of ~ 3 weeks).	IA administration of controlled-release EPA can be a new therapeutic approach to target inflammatory and catabolic markers also in patients with ObOA.	[235]
Seed oil (DSO) in niosomes	Carrageenan-induced paw oedema in rats.	0.5 g/kg DSO pure extract a day. i.p. injection.	Controlled release and therapeutic effective level of DSO niosomes in mitigating OA progression	▪ Good stability, optimum entrapment efficiency and sustained release pattern; ▪ ↓ Inflammation.	Nanoparticles as a targeted delivery system can be a valuable tool for ObOA.	[236]
Cel-MEs@MNs (microemulsion-incorporated dissolving microneedle co-loading celecoxib and α-linolenic acid)	OA model.	Transdermal injection vs. oral administration of celecoxib and α-linolenic acid.	Synergistic anti-inflammation and potent transdermal delivery,	▪ ↓ Inflammatory cytokines; ▪ ↓ Cartilage damage, paw swelling; ▪ M2 repolarization; ▪ ↓ M1 macrophages; ▪ ↓ Chondrocyte apoptosis; ▪ ↓ PGE-2.	Microemulsion with improved transdermal injection potency holds great potential in the solubilization of water-insoluble drugs.	[240]

## 5. Conclusions

In this review, we emphasized how Ob has a causative relationship with OA joint impairment not only due to altered biomechanics but also to the release of key inflammatory factors by adipose tissue. Inflammatory cytokines and adipokines may represent potential biomarkers related to ObOA, thus representing valid targets for intervening with dietary supplementation as a preventive strategy to address OA progression. We raised attention to the overall negative role of n-6 PUFAs in exacerbating degenerative features and the positive role of n-3 PUFAs in exerting anti-inflammatory, anti-catabolic and anti-nociceptive effects. We showed how the modulation of FAs could play a specific therapeutic role in ObOA by affecting the composition of gut microbiota, modulating immune systems and activating several anti-inflammatory signaling pathways, thus preserving joint degeneration. Specifically, we proposed a nutraceutical-oriented strategy focused on restoring impaired FA compositions specifically in patients with ObOA by modulating the ratio of n-6 PUFAs/n-3 PUFAs in favor of n-3 PUFAs. Notably, nanoparticle technologies offer a wide spectrum of possibilities for improving the safety and stability of n-3 PUFAs. It is difficult for scientists to guarantee the concentration of PUFAs in OA joints; therefore, this review may be useful to push forward tissue engineering approaches that include injecting PUFAs intra-articularly to overcome the current limitations of the systemic route of administration.

## Figures and Tables

**Figure 1 ijms-24-09340-f001:**
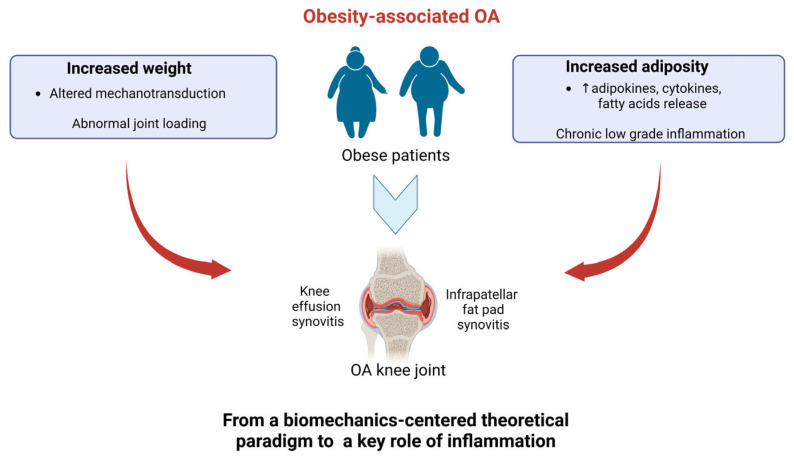
Schematic representation of the new concept in Ob-associated OA from a biomechanics-centered theoretical paradigm to a key role of inflammation. Up arrows refer to increase.

**Figure 2 ijms-24-09340-f002:**
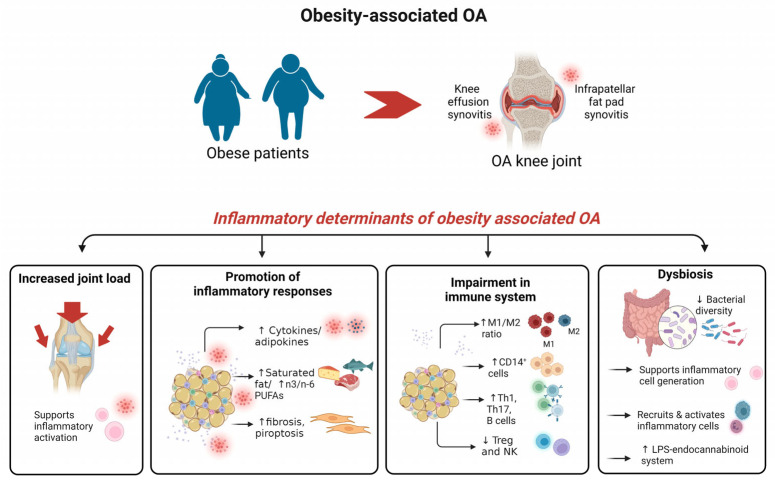
Schematic representation of biological features beyond inflammation in ObOA in the knee joint. Inflammation is among the main pathophysiological processes involved in joint degeneration and pain in ObOA. Determinants of inflammation in ObOA are activation of inflammatory mediators, impairment in the immune system and dysbiosis. Up arrows refer to increase and down arrows to decrease.

**Figure 3 ijms-24-09340-f003:**
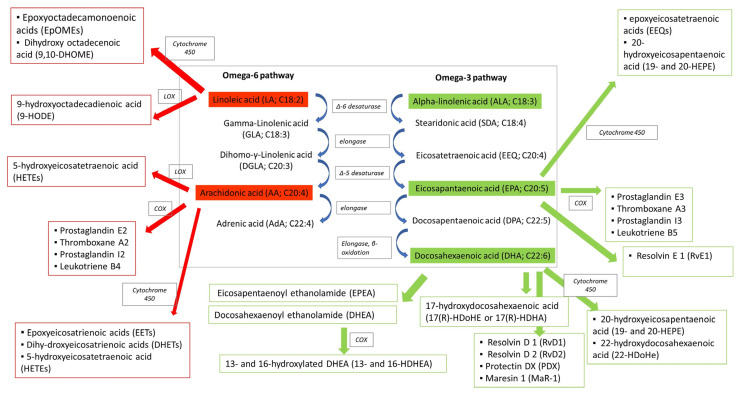
Schematic representation of signaling pathways of n-3 and n-6 PUFAs with indications of the metabolites and enzymes implicated. Dietary linoleic acid (LA, n-6 PUFAs) and alpha-linolenic acid (ALA, n-3 PUFAs) are both metabolized by the actions of elongases and desaturases to downstream key metabolites, among which are arachidonic acid (AA, n-6 PUFAs), docosahexaenoic acid (DHA, n-3 PUFAs) and eicosapentaenoic acid (EPA, n-3 PUFAs). PUFAs are mainly metabolized by cyclooxygenase (COX), lipoxygenase (LOX) and cytochrome P450 enzymes producing prostaglandin, leukotrienes and endocannabinoids. Red color highlights n-6 PUFA pathways, while green color highlights n-3 PUFA pathway.

**Figure 4 ijms-24-09340-f004:**
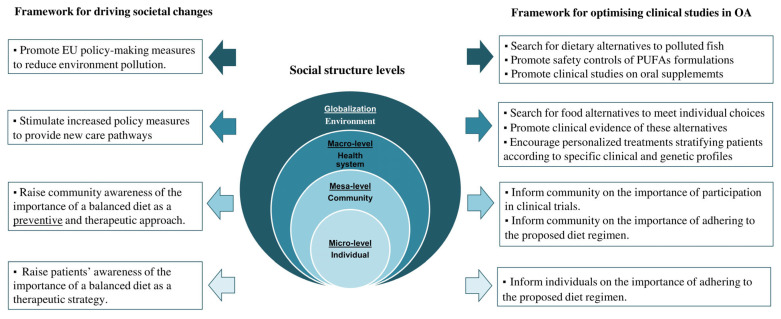
Levels of social structure and framework to optimize clinical studies in OA. There is an urgent need to well inform individuals and the community to strictly adhere to the proposed diet regimen, to ensure more rigorous studies and avoid the possibility of risk bias in interpreting clinical results. The health system needs synergistic cooperation between health professionals to identify novel food alternatives and develop personalized medicine through the use and integration of different analyses. Indeed, on global level, more safety controls on PUFA formulations, searches for dietary alternatives rich in n-3 PUFAs and more rigorous clinical studies are necessary.

**Figure 5 ijms-24-09340-f005:**
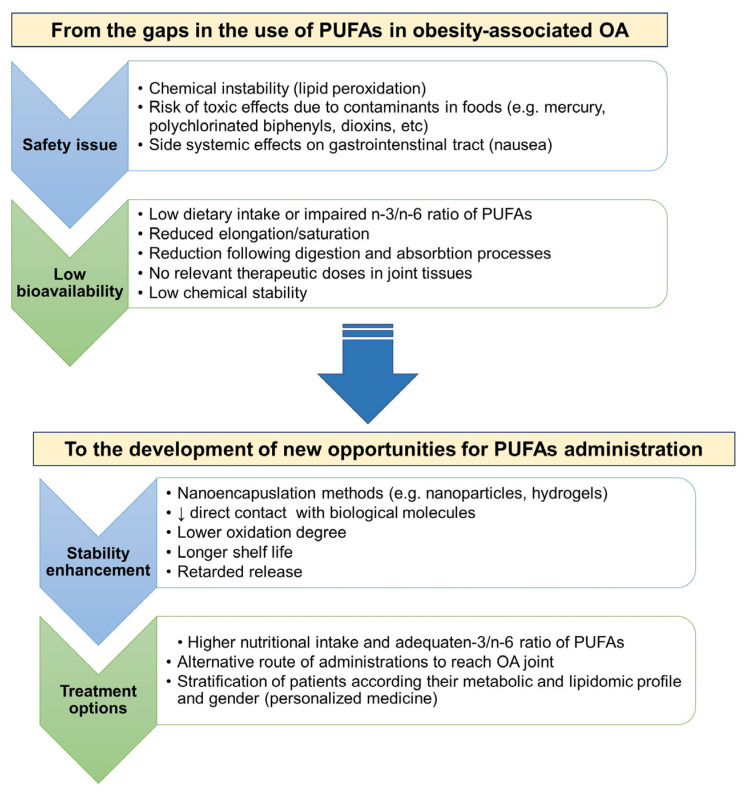
Schematic representation of measures to close gaps and create new opportunities in using polyunsaturated fatty acids in Ob-associated OA. Safety and lack of availability are the major limitations of using these phytochemicals. Safety issues mainly include aspects of chemical instability that cause the generation of toxic products, both systemically and locally. The presence of contaminants in n-3 PUFA-rich foods predisposes various side effects. New opportunities to optimize the use of PUFAs lie in improving their stability and establishing new treatment options to improve clinical success rates.

**Table 2 ijms-24-09340-t002:** Biological effects of Ob and PUFAs (especially n-3 PUFAs) in preclinical in vivo models. Up arrows refer to increase and down arrows to decrease.

Molecule	In Vivo Model	Dose/Delivery Route	Main Effects	Specific Outcomes	Indications for Preventive-Therapeutic Strategies for ObOA	Ref.
N/A	Post-traumatic model of OA: ACLT model in rabbits	N/A	Early-stage OA affected FA composition towards a pro-inflammatory phenotype.	▪ ↓ n-3/n-6 PUFA ratio in the IFP.	Importance of preventing n-3/n-6 PUFAs imbalance/restoring n-3/n-6 PUFAs balance.	[132]
HFD (60% kcal fat)	Ob model: male C57BL/6J mice + HFD	- *Group 1*: control (10% kcal fat) - *Group 2*: HFD (60% kcal fat) Dietary supplementation	HFD caused moderate OA.	▪ ↑ Association with systemic biomarkers, e.g., IL-1β, glucose intolerance (metabolomic analysis); ▪ ↑ Cartilage destruction (Mankin score); ▪ ↑ BMD.	Avoiding HFD to prevent OA onset or delay OA aggravation.	[139]
MFD	Ob model: male C57BL/6J (B6) mice + MFD	- *Group 1*: control diet (4% fat; d omega-3/omega-6 ratio: 1.29) - *Group 2:* “Ob-induced diet” with medium fat content (11% fat); n-3/n-6 PUFA ratio: 3.48 for 24 weeks.	MFD promoted changes in immune metabolism and altered gut microbiota composition.	▪ ↑ Leptin; ▪ Altered gut microbiota composition; ▪ Modulated the immune metabolic response of adipose tissue.	Fat diet content above 11% induces metabolic changes.	[140]
HFD (60% kcal fat)	Ob model: male C57BL/6J mice + HFD	- *Group 1*: HFD - *Group 2*: control exercised (aerobic exercise: running wheel exercise) or sedentary mice Dietary supplementation	HFD promoted OA onset. Moderate exercise improved glucose tolerance without reducing body fat or cytokine levels.	HFD promoted OA onset by: ▪ Increasing serum levels of leptin, adiponectin, KC (mouse analogue of IL-8), MIG and IL-1RN ▪ Decreasing glucose clearance from the blood; ▪ Increasing subchondral bone thickness and GAG loss; ▪ Running wheel exercise produced modest changes in knee histopathology.	Avoid HFD for preventing OA onset or delaying OA aggravation. Recommend aerobic exercise in promoting joint health independently of weight loss.	[137]
PUFAs diet	ObOA model: DMM model in male mice + HFD (60% kcal fat)	- *Group 1*: HFD rich in n-3 PUFAs - *Group 2*: HFD rich in n-6 PUFAs - *Group 2*: HFD rich in SFAs - *Group 2*: control (10% kcal fat) Dietary supplementation	Protective role of dietary supplementation with n-3 PUFAs in mitigating OA changes was observed.	▪ Negative correlation of serum n-3 PUFAs with OA; ▪ Positive correlation of serum n-3 PUFAs with adiponectin and priming to M2 macrophages; ▪ Positive correlation of serum n-6 PUFAs with OA and inflammatory adipokines.	Shifting the composition of Fas in the diet towards a low intake of n-6 PUFAs and SFAs and a high intake of n-3 PUFAs for mitigating OA.	[15]
PUFAs diet	ObOA model: DMM model in male mice + HFD	- *Group 1*: HFD + SFA - *Group 2*: n-3 PUFAs (8% by Kal) - *Group 3*: n-6 PUFAs Dietary supplementation	Dietary FA content modulated OA severity; small amounts of n-3 PUFAs could mitigate OA while independently increasing OA severity.	▪ n-3 PUFAs reduced cartilage degradation synovitis and macrophage infiltration; ▪ n-6 PUFAs increased cartilage degradation and bone erosion and exhibited synovitis and infiltrating cells; ▪ SFA and n-6 PUFAs increased leptin concentrations; ▪ n- 3 PUFAs increased adiponectin levels and reduced leptin and resistin levels.	Shifting of FA composition in diet toward a low intake of n-6 PUFAs and SFAs and a high intake of n-3 PUFAs for mitigating OA.	[141]
HFD rich in n-6 PUFAs or control diet	ObOA model: DMM model + HFD in male and female fat-1 transgenic mice (encoding n-3 PUFAs desaturase)	- *Group 1*: fat-1 mice - *Group 2*: wild-type mice Dietary supplementation	Circulating FA composition and metabolic inflammation rather than “mechanical” factors were major risk factors for ObOA.	▪ Conversion to n-3 PUFAs reduced OA and synovitis in a sex- and diet-dependent manner; ▪ ↓ Serum pro-inflammatory cytokines; ▪ ↑ Anti-inflammatory cytokines; ▪ No changes in body weight.	Potential genetic use of ω-3 FA desaturase to reduce ObOA.	[142]
Standard diet	Obese model of OA: DMM model in fat-1 transgenic (TG) mice vs. Wild-type mice	- *Group 1*: fat-1 transgenic mice - *Group 2*: wild-type mice	Protective role of desaturase in mitigating OA, probably through inhibition of mTORC1, and the promotion of autophagy and cell survival in cartilage chondrocytes was observed.	▪ TG mice increased levels ▪ of n-3 PUFAs; ▪ ↓ MMP-13, ADAMTS; ▪ ↓ Cartilage destruction and osteophyte formation; ▪ ↓ TORC-1 activity; ▪ ↑ chondrocyte autophagy.	Potential genetic use of n-3 PUFAs desaturase to reduce ObOA.	[143]
Diet supplemented with 10% safflower oil and with 23% kcal fat, with an n-6:n-3 ratio of 274	Spontaneous OA model in fat-1 TG mice	- *Group 1*: fat-1 transgenic mice - *Group 2*: wild-type mice Life-long reduction	Cartilage degeneration and osteophyte formation was developed at levels comparable to WT mice.	▪ The n-6:n-3 ratio was reduced twelve-fold in males and seven-fold and females; ▪ no reduction in cartilage, synovium or bone-associated OA changes; ▪ modest reduction in IL-6 and TNF-α levels.	Lower efficacy of n-3 PUFAs desaturase in spontaneous OA than post-traumatic OA models.	[144]
No diet	Ob models: leptin-deficient (ob/ob) and leptin-receptor-deficient (db/db) female mice No OA induction	N/A	Impaired leptin signaling significantly altered subchondral bone morphology without altering knee OA.	▪ Increased body mass and fat; ▪ Reduced subchondral bone thickness; ▪ Increased relative trabecular bone volume in the tibial epiphysis; ▪ No cartilage degeneration; ▪ No changes in systemic inflammatory cytokines.	Leptin signaling is key to inducing systemic inflammation.	[145]
ALA, EPA, DHA	Ob models: - C57BL/6J mice + HFD (236 g/kg fat); - leptin-deficient (ob/ob) + HFD (236 g/kg fat)	C57BL/6J 16 weeks ob/ob mice for 6 weeks	Supplementation with EPA, but not ALA and DHA, could preserve glucose homeostasis in an obesogenic environment and limit fat mass accumulation.	HFD: ▪ Three PUFAs incorporated into erythrocyte PLs (EPA, DHA > ALA); ▪ ↑ EPA, DHA in adipose tissue; ▪ ↓ Plasma cholesterol (EPA); ▪ ↓ Fat mass accumulation (EPA); ▪ ↓ Adipose cell hypertrophy (EPA); ▪ ↓ Insulin sensitivity and glucose tolerance. Ob/ob mice ▪ Partial protection against glucose intolerance and IR (EPA); ▪ ↑ Adiponectin; ▪ ↑ Phosphorylation of Akt.	EPA is more effective in targeting specific Ob features.	[146]
n-3 PUFAs	Spontaneous model of OA (OA guinea pig)	- *Group 1*: n-3 PUFAs - *Group 2*: standard diet Dietary supplementation	Chondroprotective effects were observed.	▪ ↓ Cartilage score; ▪ ↑ GAG content; ▪ ↓ Denatured Coll II; ▪ ↓ MMP-2.	Relevance of a diet rich in n-3 PUFAs to counteract cartilage degradation.	[147]
GLM abundant in DHA	MIA-induced OA model in male Wistar rats	GLM (100–300 mg/kg) versus celecoxib (50 mg/kg) Oral administration (3 days from MIA injection)	Chondroprotective properties and a reduction in catabolic, inflammatory and necroptotic markers were observed.	▪ ↓ Pain; ▪ ↓ Cartilage destruction; ▪ ↓ T and B cell responses; ▪ ↓ MMP-1/-3/-13 in cartilage; ▪ ↓ IL-1β, IL-6, NF-KB, i-NOS in synovium; ▪ ↓ Necroptosis-related markers (RIPK1, RIPK3, pMLKL).	Potential candidate in targeting inflammation and necroptosis.	[115]
PDX (DHA metabolite)	MIA-induced OA model in Sprague–Dawley rats	10 µg/kg (every 3 days) Intraperitoneal injections	Chondroprotective and anti-inflammatory effects were observed.	▪ ↓ Cartilage degradation; ▪ ↑ Cartilage thickness and cartilage surface; ▪ ↓ TNF-α in the serum and intra-articular lavage fluid.	Potential tool to target inflammatory hallmarks.	[118]
DHA	Post-traumatic OA: ACLT-induced OA model in male Sprague–Dawley rats	1 mg/kg (two months) Injection in tail vein	Promotion of bone remodelling and cartilage reduction were observed.	▪ ↓ TRAP, RANKL, CD31, endomucin agents (markers of bone loss); ▪ ↓ OARSI score; ▪ ↓ MMP-13, collagen X.	Potential of targeting catabolic markers.	[148]
Antarctic krill oil (*Euphausia superba*) (rich in EPA and DHA)	DMM-induced OA in osteoporotic (ovariectomy) mice	Diet supplementation	Chondroprotection and reduction in inflammation were observed.	▪ ↓ NF-κB pathway through the activation of GPR120; ▪ ↓ Cartilage degeneration.	Potential of targeting inflammatory markers.	[134]
Triglyceride n-3 oil (rich in DHA + EPA)	Naturally occurring OA in dogs Prospective, randomized, double-blind, placebo-controlled clinical trial.	69 mg EPA + DHA/kg/day (84 days) Diet supplementation	Improvement in clinical markers of OA was observed.	▪ ↓ AA in blood; ▪ ↓ Effusion and pain from day 42.	Potential of reducing systemic inflammation.	[149]
EPA and DHA	OA horses	Diet supplementation	Increased storage pools of n-3 PUFAs in SF and anti-inflammatory effect	▪ ↑ n-3 PUFA in the SF; ▪ ↑ Surfactant glycerophosphocholines (GPC); ▪ ↓ Inflammation.	Potential to improve the resolution of inflammation.	[150]
17^®^-HDoHE (RvD2 precursor)	MIA model of OA + MNX model of OA	1 ng/μL (every day from 14 to 28 days) Intra-peritoneal administration	Long-term inhibitory effects on nociceptive signaling.	▪ ↓ Pain; ▪ ↓ Astrogliosis in the spinal cord; ▪ ↑ RvD2 in plasma.	Potential to exert analgesic potential.	[151]
Aspirin-triggered RvD1 (AT-RvD1) RvD1 isomer induced by aspirin and more resistant to enzymatic degradation than RvD1	Carrageenan-induced inflammatory or MIA-induced OA in male Sprague–Dawley rats	15 ng in 50 μL PBS (carrageenan model) 15 ng and 150 ng in 50 μL PBS (MIA model) Spinal treatment	The selective target of inflammation drives spinal hyperexcitability in nociceptive pathways (analgesic potential)	▪ ↓ Peripheral nociceptive fiber-evoked responses; ▪ ↑ ChemR23; ▪ ↑ FPR2/ALX; ▪ ↓ NMDA receptor activation.	Potential to exert analgesic potential.	[152]
RvE1/RvD1/PDX	Rat paws inflamed by carrageenan or histamine, 5-hydroxytryptamine, substance P or prostaglandin E2	20 RvE1, 100 RvD1,100 μg PDX mL^−1^ versus standard anti-inflammatory drugs (INDO, celecoxib and dexamethasone) Injection in the hind paws (10 min before the stimuli)	Analgesic and anti-inflammatory effects	▪ ↓ Inflammation (RvE1, RvD1); ▪ Analgesic effects (RvE1, RvD1); ▪ Cuperior effect of RvE1 than RvD1.	Rvs, as analgesic agents, may be a better therapeutic agent than NSAIDs.	[153]
MaR2	Lipopolysaccharide (LPS)-induced mechanical hyperalgesia capsaicin (TRPV1 agonist) or AITC (TRPA1 agonist).	3, 10 or 30 ng	Analgesic effect	▪ ↓ Cytokine; ▪ ↓ TRPV1, TRPA1 activation.	Potential analgesic effects.	[154]
Linseed oil (LO), soybean oil (SO) and peanut oil (PO) n-6/n-3 PUFA ratios: 1:3.85 (LO), 9.15:1 (SO) and 372.73:1 (PO)	DMM OA murine model	12 weeks Oral supplementation	Edible oils with low n-6/n-3 PUFAs exert an anti-inflammatory effect by inhibiting the NFκB pathway.	▪ ↑ Cartilage thickness (LO, SO); ▪ ↓ TNF-α in serum (LO, SO); ▪ LO or SO activated GPR120 and attenuated EP4 (LO, SO); ▪ ↓ NFκB pathway; ▪ ↓ MMP-13, ADAMTS-5.	Anti-inflammatory potential of a low-n-6/n-3-PUFA diet.	[155]

## Data Availability

Not applicable.

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
