# Peer review of "Targeting the Inflammatory Hallmarks of Obesity-Associated Osteoarthritis: Towards Nutraceutical-Oriented Preventive and Complementary Therapeutic Strategies Based on n-3 Polyunsaturated Fatty Acids"

_ijms, 2023, doi:10.3390/ijms24119340_

Round 1

Reviewer 1 Report

1. Several researchers have reviewed recent advances in the treatment of Osteoarthritis with PUFAs in recent years, and this article does not present a more original perspective, which would lead to its lack of innovation.(doi: 10.1016/j.plipres.2021.101113. Epub 2021 Jul 2; doi: 10.1155/2021/4878562. eCollection 2021).

2.The authors attempted to focus on obesity-associated osteoarthritis and summarized its onset by mechanisms such as increased joint load, inflammatory responses, impairment of the immune system and dysbiosis. However, the subsequent discussion focused only on n-3 PUFA and inflammatory response. This seems to be insufficient to adequately establish a direct relationship between n-3 PUFA and obesity-associated osteoarthritis. This review essentially reviewed the relationship between n-3 PUFA and osteoarthritis in general.

3. The format of inserting references in the article is not standardized and needs to be revised.

Reviewer 2 Report

The authors have reviewed the literature on nutraceutical-based therapies for obesity-induced osteoarthritis. The manuscript outlines obesity induced osteoarthritis and the underlying mechanisms for its pathogenesis. Specifically, changing obese patient diet from n-6 polyunsaturated fats to n-3 polyunsaturated fats can help reduce OA progression.

1)    Tables in the manuscript need to be reformatted – up and down arrows are difficult to observe and formats need to be consistent between tables.

2)  References need to reformatted and not single references for each article.

3)     In respect to personal diet, although there is a focus on Western diet in the publication (section 4.2), how does ω-6 and ω-3 balance in other diets (e.g. Asian and African) influence their susceptibility to osteoarthritis ? A comment in the review and potential methods to help to prevent osteoarthritis in those populations.

4)     The review is focused on lipid metabolism but how do other nutritional elements (i.e. carbohydrates, proteins) influence ω-6 to ω-3 conversion and are these nutrients required in balance. It is also known that carbohydrate to fat conversion is a contributor to osteoarthritis. A comment on this part or a section should be included in this review.

5)      In the studies examining the use of ω-3 administration for treating OA in human studies, what was the level of OA prior to administration and were certain groups having significant improvements upon ω-3 treatment in their KOOS, WOMAC or VAS score ? There is a consensus that OA treatment requires a stratified approach and so do these studies state differences in responders and non-responder patients to ω-3 treatments.

Reviewer 3 Report

Peer Review IJMS - 2316315

The review manuscript entitled “A nutraceutical-based therapeutic strategy for obesity-associated osteoarthritis: from current evidence to translational perspectives for n-3 polyunsaturated fatty acids administration” from Laura Gambari et al. provides and updated and well-organized revision of the current evidence on the anti-inflammatory effects of n-3 polyunsaturated fatty acids with particular focus for applications in the treatment of obesity-associated osteoarthritis. The topic of this review is highly relevant and within the scope of the International Journal of Molecular Sciences (ISSN 1422-0067). Moreover, it is particular suited for the Special issue “New Frontiers in Musculoskeletal Tissue Repair and Tissue Regeneration”. However, there are some minor issues that should be addressed before being considered for publication (Accept after Minor Revision).

Comments:

1.     (line 77, Figure 1 caption): Please remove the ”a” to obtain “(...) theoretical paradigm(...)”.

2.     (lines 177 and 178). Please standardize the use of TRPV1 or TRPV-1.

3.     (line 270): It appears that a word is missing in the sentence. Please correct to improve clarity.

4.     (line 342): The authors probably mean Mar-1 instead of Mar-11. Please correct this issue.

5.     (Table 1): Please improve the Table format. The words/sentences from different table columns are too close and difficult to read. Same issue is valid for Table 2.

6.     (line 371): Please add a space to obtain:”(...) to 1989[108].”

7.     (line 425): It is better to replace “for” by “to” to obtain: “(...) a drug used to treat OA [121].”

8.     (lines 433-435): Please add commas to the sentence to improve clarity.

9.     (line 438): Please correct the verb form “has” to “have” to obtain: “(...) OA has the dietary control (...)”

10.  (Table 3, last entry, 2nd column- study types): Please correct “16andomized” to “randomized”.

11.  (lines 467/468): It appears that a word is missing. The word “relevant” should be added to obtain: “(...) to reach therapeutically relevant concentrations (...)”.

12.  (line 479): Please erase the misplaced final point to obtain: “(...) DHA [134].”

13.  (lines 596-599): The sentence needs to be rewritten/improved.

14.   (lines 626-627): The terms “improved cartilage damage” appears wrong in the context of the statement. Please revise it.

15.  (line 641): Please correct the verb form “plays” to “play”.

16.  (line 650): Please correct to the singular form to obtain: “(...) we proposed therapeutic interventions based on (...)”.

17.  (line 652): Please replace “how” by “that” to achieve the desired meaning of the sentence.

18.  (line 659/660): Please improve the last sentence to avoid the repetition of the word “administration”.

19.  If possible, the authors should present some examples of tissue engineering strategies combined with n3-PUFA to promote cartilage regeneration and decrease joint inflammation.

Round 2

Reviewer 1 Report

The authors have responded all comments and reviesed the manuscript accordingly. Please consider whether the length of the article is too long, and English writing should be further improved.

Reviewer 2 Report

The authors have answered my questions appropriately.